# Measuring gas discharge in contact electrification

Hongcheng Tao ●[1] & James Gibert ●[1] ✉

Contact electrification in a gas medium is usually followed by partial surface charge dissipation caused by dielectric breakdown of the gas triggered during separation of the surfaces. It is widely assumed that such discharge obeys the classical Paschen's law, which describes the general dependence of the breakdown voltage on the product of gas pressure and gap distance. However, quantification of this relationship in contact electrification involving insulators is impeded by challenges in nondestructive in situ measurement of the gap voltage. The present work implements an electrode-free strategy for capturing discrete discharge events by monitoring the gap voltage via Coulomb force, providing experimental evidence of Paschen curves governing nitrogen breakdown in silicone-acrylic and copper-nylon contact electrification. It offers an alternative approach for characterizing either the ionization energies of gases or the secondary-electron-emission properties of surfaces without the requirement of a power supply, which can potentially benefit applications ranging from the design of insulative materials to the development of triboelectric sensors and generators.

Contact electrification, the intriguing natural phenomenon of electric charge transfer between touching surfaces, has been studied for centuries. The underlying charging mechanism, however, remains under debate partly due to challenges in quantifying the resultant surface charge density[1–4] which is potentially hindered by a stage of discharge during surface separation (Fig. 1a). At an infinitesimal gap immediately after disengaging, the surfaces possess a raw amount of opposite charge. When they continue to separate, the surface charge forms an electric field across the gap which is subsequently filled by any gaseous or liquid medium that flows in from the surroundings. As the gap voltage increases with distance, it may trigger dielectric breakdown of the medium and thus partially dissipate the surface charge[5–8]. In atmospheric air, the first breakdown events usually happen within a few micrometers, thus concealing the initial charge density. In real life, while most often noticed as little shocks from a winter laundry, sparks generated by surface charge may pose fire and explosion hazards in dairy farms as well as in industrial processes involving powders and fabrics[9–11]. On the contrary, the lack of gas discharge in space instead causes insulative parts in satellites to break down from heavy surface charge buildup[12,13]. Meanwhile, succeeding research in electrostatic

generators dating back to the 1700s[14], the significance of gas breakdown is also acknowledged recently in energy harvesters that employ contact electrification, namely triboelectric generators[15,16], where it can be either a limiting factor of output performance[17–19] or instead exploited as a mechanism of current[20,21]. A comprehensive model of the gas breakdown discharge process in contact electrification is therefore desired in these scenarios and has conventionally been based on Paschen's law[22,23] which describes the dependence of breakdown voltage $V_b$ on the product of gas pressure $p$ and gap distance $d$ as

$$V_b = \frac{Bpd}{\ln(Apd) - \ln[\ln(1 + \gamma_{se}^{-1})]} \quad (1)$$

where constants $A$ and $B$ are determined by the gas constituents, and the secondary-electron-emission coefficient $\gamma_{se}$ is also dependent on the surface materials. While Paschen's law has been widely assessed for gas discharge between electrodes with a high voltage power supply, its applicability to gas breakdown triggered by finite surface charge due to contact electrification, especially between insulators, has

[1]School of Mechanical Engineering, Purdue University, West Lafayette, IN, USA. ✉e-mail: jgibert@purdue.edu

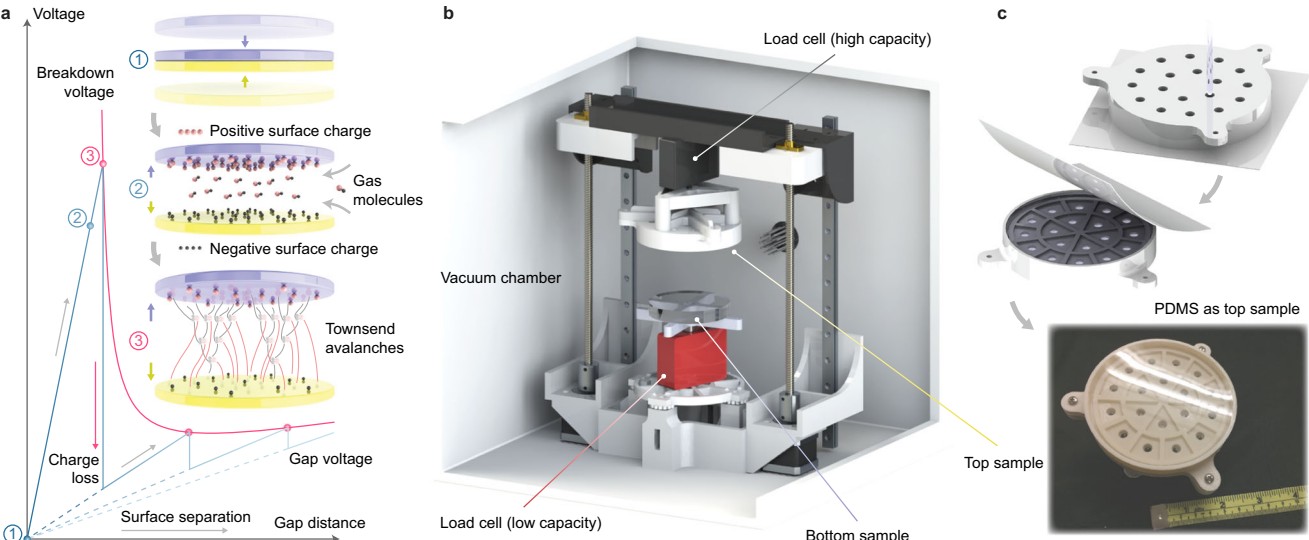

**Fig. 1 | Phenomenon description and test setup. a** Stages of a typical contact electrification cycle between insulators in a homogeneous gas with constant pressure: (1) Electrically neutral surfaces forced into intimate contact. (2) Surface separation immediately after disengaging, where the raw amount of surface charge (black being negative, red positive) is maintained while the gap is filled with gas molecules (black being electrons, red cations) from the surrounding. (3) The first gas breakdown event as the gap voltage approaches the breakdown threshold. Discharge occurs in the form of self-sustaining cascades of Townsend avalanches, partially reducing the surface charge density and thus the gap voltage. More breakdown events follow as the gap increases. **b** Test apparatus in an acrylic vacuum chamber, where motion of the top sample is driven by stepper motors and surface alignment is calibrated by leveling screws under the bottom sample. **c** Fabrication of a PDMS (polydimethylsiloxane) surface as top sample, where liquid PDMS is cast in a 3D-printed plastic mold pressed against a clear polyester sheet on a flat surface.

insufficient experimental validation. Difficulties lie in monitoring the gap voltage in situ during surface separation since the placement of electrodes connected to an external circuit may disturb the electric field by induced charge, while the electrodes' geometry and location may affect the accuracy of voltage measurement, regardless of surface conductivity. At the same time, the measurement of gas breakdown voltage also requires both a range typically exceeding 1 kV and a high input impedance. The present work therefore implements an alternative nondestructive approach similar to setups reported in prior works[7,24] which uses Coulomb force measurements to monitor surface charge variations and thus quantify the breakdown voltage of a gas medium between electrified surfaces with respect to its pressure and the gap distance. It is illustrated with the reconstruction of complete Paschen curves for nitrogen breakdown in both silicone-acrylic and copper-nylon contact electrification.

## Results

### Experimental approach

The test apparatus (Fig. 1b) performs contact electrification in a vacuum chamber and thereafter measures the attractive Coulomb force between the charged surfaces when they are separated. A load cell with high capacity (25 N) is mounted above the top sample surface to monitor the contact force as well as any strong adhesion when the surfaces disengage, the majority of which is attributed to van der Waals interactions[25]. A second load cell with low capacity (1.2 N) is placed beneath the bottom sample surface to measure the Coulomb force, which is overloaded during the contacts as a compromise. The top and bottom sample surfaces are planar with a circular effective contact area (45.6 cm², 76.2 mm diameter) which is relatively large to ensure sufficient load cell resolution for capturing low-voltage gas break-down. Two test strategies, namely pseudo-constant-pressure and pseudo-constant-distance tests, are implemented to reconstruct the presumed Paschen curve by detecting gas breakdown events when gap distance and gas pressure are varied, respectively. A pseudo-constant-pressure test simulates the general contact electrification process by separating charged surfaces at different controlled gas pressures

(Fig. 2a). The vacuum chamber is first flushed with the operating gas, where the surfaces are brought to a significant gap distance around 30 mm while the gas pressure is swept between 10 Pa and 100 kPa 3 times. It is assumed that the majority of any residual surface charge is dissipated during this stage, at which point the load cells are zeroed. The gap is then slowly closed until a contact force is detected and the corresponding displacement is recorded as the nominal zero point for measuring gap distance. The gas pressure in the chamber is then lowered and kept around 10 Pa, where the surfaces are pressed into several quasi-static contact cycles with a controlled peak contact force until a certain amount of surface charge is deposited. The surface charge density is in general not saturated but assumed uniform, while the maximum gap distance in the separation stage of each contact cycle is kept small (less than 2 mm) to avoid triggering gas breakdown, albeit ideal disengaging at exactly zero gap distance is usually not feasible since extra tension is required to overcome any van der Waals adhesion. After the surfaces fully disengage at the end of the final contact cycle, they are brought back to a near-zero gap (around 0.02 mm) and the gas pressure is raised and kept closely around a target value. The gap is then increased quasi-statically, during which the Coulomb attraction is monitored and post-processed to calculate the surface charge density and gap voltage. Each discontinuity (drop) in the measured voltage represents a breakdown event and is recorded as an intersection with the hypothetical Paschen curve. The first breakdown event in tests with comparatively high initial raw charge density reveals earlier sections (at smaller gaps) of the Paschen curve, and ideally the further parts (at larger gaps) can later be covered by breakdown events that follow. In practice, the lowest measurable voltage is limited by the resolution of the load cell so that tests at various target gas pressures are performed to obtain different sections of the Paschen curve. When the target gas pressure is high, massive discharge may already happen while the pressure is being raised to the target value, in which case an alternative strategy (path 2, Fig. 2a) is employed where the contact cycles are performed at a higher gas pressure (e.g., atmospheric, around 100 kPa) instead. The separation stage in the contact cycles is no longer free of discharge but an

adequate amount of residual surface charge can still be deposited[26]. Similarly, the surfaces are brought to a near-zero gap after the final contact cycle and gas pressure is then lowered to the target value, followed by the same separation and Coulomb force measurement procedure.

A pseudo-constant-distance test instead fixes the gap length and records the breakdown events as the gas pressure varies (Fig. 2b). Similarly, contact electrification cycles are first performed at a low pressure around 10 Pa. At the end of the final contact cycle, the surfaces are moved to the target gap distance and the operating gas is slowly released into the chamber while the Coulomb force is monitored so that breakdown events are indicated by discontinuities in the measurement. However, discharge detection by increasing the gas pressure may only reveal the first half (left to the minimum voltage point) of the Paschen curve since intersections with the second half is not feasible once the gap voltage falls below the minimum breakdown threshold. An alternative strategy (path 2, Fig. 2b) starts with contact electrification cycles at a high gas pressure (e.g., atmospheric, around 100 kPa) and then slowly pumps gas out of the chamber while the surfaces are kept at the target gap distance. Breakdown events represented by intersections with the second half of the Paschen curve can hence be obtained.

## Nitrogen discharge in silicone-acrylic electrification

Test results of nitrogen discharge between PDMS (polydimethylsiloxane, top) and acrylic (PMMA, polymethyl methacrylate, bottom) surfaces (charged under contact cycles with a peak force of 24 N, 5.2 kPa) at room temperature 20 °C are depicted in Fig. 3, where Fig. 3a, b demonstrate the gap voltage monitored during multiple test runs and the detected breakdown events are collected in Fig. 3c. The test conditions have been described as pseudo-constantly controlled

since in each pseudo-constant-pressure test the gas pressure has minor fluctuations around the target value, as labeled, while in both test strategies the true gap distance is subject to deflections of the load cells (Supplementary Movie 1) and consequently a discharge event causes a reduction of Coulomb force and therefore an increase of gap distance, which is compensated for in post-processing. In the same plots the theoretical Paschen curves for nitrogen[22] with coefficients $A = 8.85 \, Pa^{-1} \, m^{-1}$ and $B = 243.77 \, VPa^{-1} \, m^{-1}$ are displayed assuming two reference values 0.3 and 0.005 for $\gamma_{se}$, since secondary-electron-emission properties of the tested PDMS surface (PDMS gains negative charge against acrylic - the polarity is determined by direct surface charge collection in room air using a brush electrode grounded through an electrometer, explained in Supplementary Method 1 and Supplementary Fig. 1) under bombardment of nitrogen cations remain to be characterized. The assumption that $\gamma_{se}$ is invariant is challenged since the probability of secondary electron emission from the insulator surface generally increases with the energy possessed by the incident cations[27,28]. Assuming $A$ and $B$ are constant, effective values of $\gamma_{se}$ are calculated by plugging gap voltage, gas pressure and gap distance measurements at each breakdown event in Fig. 3c into Paschen's law (1), as plotted in Fig. 3d with respect to the reduced electric field $V_b/(pd)$ which is theoretically proportional to the average energy of incident cations. It shows that $\gamma_{se}$ increases roughly with the reduced electric field beyond around $200 \, VPa^{-1} \, m^{-1}$, while below this level high $\gamma_{se}$ values are again observed which matches reports in literature[29-32]. The increase of $\gamma_{se}$ at low incident cation energies is attributed to secondary electron emission caused by agents other than gas cations, such as photons and metastable gas molecules resulting from non-ionizing collisions between electrons and gas molecules, as illustrated in Supplementary Discussion 1 and Supplementary Fig. 2. At low reduced electric fields, a greater portion of the kinetic energy that each electron gains from the

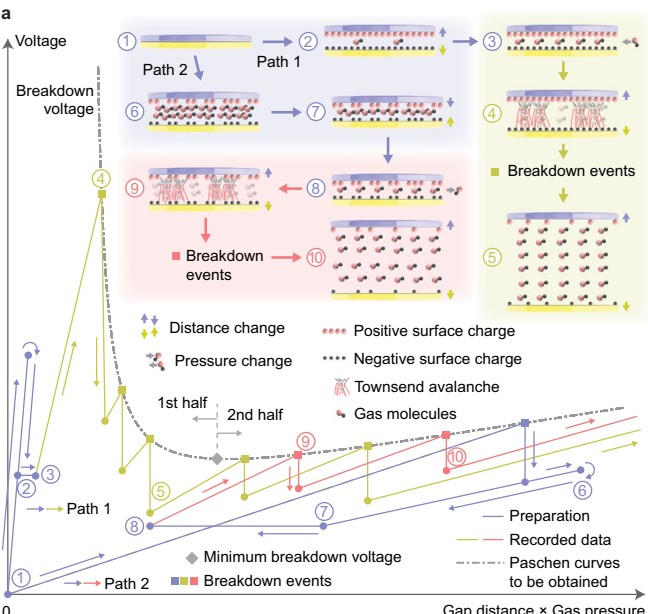
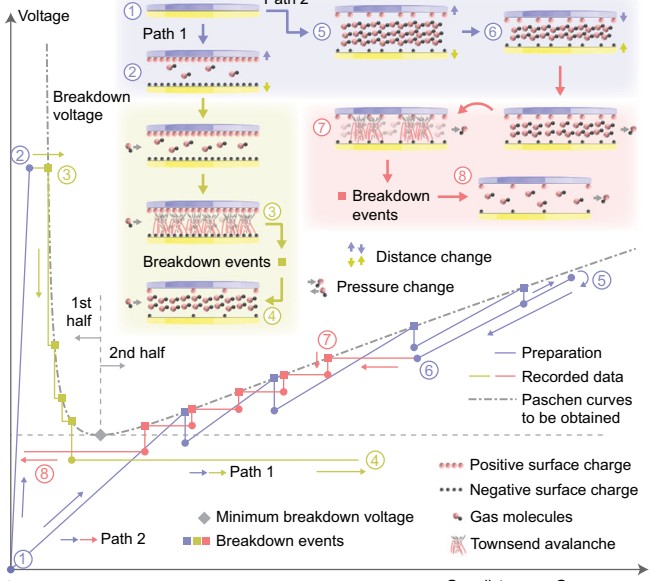

**Fig. 2 | Test strategies of measuring gas breakdown voltage in contact electrification. a** Pseudo-constant-pressure and **b** Pseudo-constant-distance. In (**a**) the gap between charged surfaces is increased at different target gas pressures (low in path 1, high in path 2) to trigger gas breakdown. In (**b**) gas pressure is varied (increasing in path 1, decreasing in path 2) at fixed gap distances to trigger gas breakdown. Schematics are provided for representative numbered states, where black spheres represent negative surface charge or electrons in gas molecules, and red spheres are positive surface charge or gas cations. Transitions between states include change in gas pressure at fixed distance (zero slope), change in gap distance at constant pressure (non-zero finite slope) and gas breakdown events

(vertical drops) indicating intersections with hypothetical Paschen curves (dashed, neutral gray) to be obtained. The contact electrification (state 1) practically involves multiple contact cycles to deposit an adequate amount of surface charge, and state 1 represents the contact phase in the final cycle. In both strategies, path 1 starts with contact electrification under low gas pressure, while path 2 starts with contact cycles under high gas pressure where partial breakdown discharge is inevitably present. In (**a**) from state 1 to 2 the gap is first raised to overcome van der Waals adhesion and then reduced, which may also apply to (**b**) depending on the target gap length at state 2.

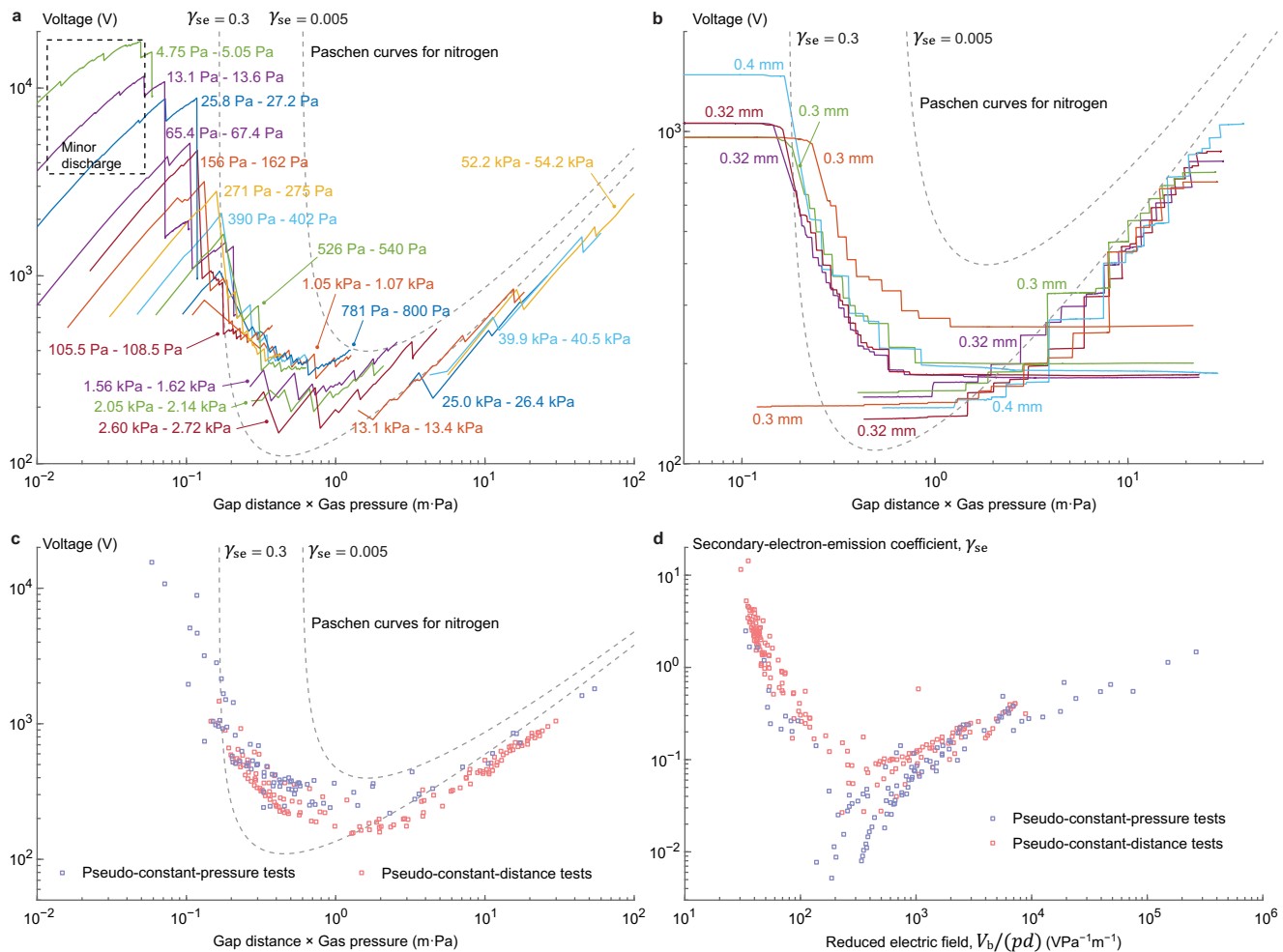

**Fig. 3 | Nitrogen breakdown in PDMS-acrylic contact electrification. a** Pseudo-constant-pressure test results at multiple target gas pressures as labeled (in color) with tolerances. **b** Pseudo-constant-distance test results at multiple target gap distances labeled (in color) with their nominal values at zero load cell deflection. **c** Collected gap voltage readings at breakdown events detected from test runs in both strategies (lavender are pseudo-constant-pressure tests results, coral pseudo-constant-distance). Dashed lines are reference curves evaluated using the classical Paschen's law (1) with empirical coefficients $A$ and $B$ for nitrogen while assuming two hypothetical values of $\gamma_{se}$. **d** Effective secondary-electron-emission coefficient of PDMS in nitrogen discharge. Each value is estimated from test results in (**c**) by plugging gap voltage, gas pressure and gap distance at each measured breakdown event into classical Paschen's law (1).

electric field is devoted to the creation of metastable gas molecules and photons instead of cations, thus adding to the effective secondary electron emission accounted for an incident cation on the negatively charged surface (PDMS). Meanwhile, the $\gamma_{se}$ values at low reduced electric fields are mostly calculated from gas breakdown events at small gaps, which reduces the diffusion of such metastable molecules and photons into the surroundings since unlike cations they are not accelerated along the electric field. Other than the deviations in $\gamma_{se}$, discrepancies are observed in the overlapping of pseudo-constant-pressure test results where the minimum breakdown voltage appears higher in tests at low target gas pressures, which is attributed to the reduced validity of the infinite-parallel-plate assumption in calculating the gap voltage as the distance increases. At the same time, it is also expected that voltage evaluation upon the infinite-parallel-plate and uniform-surface-charge assumptions result in the recorded breakdown voltages being lower than the actual Paschen curve since, unlike between electrodes, gas breakdown is always initiated at locations on the insulator surfaces with the highest voltage (and probably thereafter propagated over the entire area) which is above the calculated average. Besides, since all breakdown events between surfaces with finite charge occur as surges, they are theoretically triggered at a voltage lower than

what sustains a continuous current across the gap as in the case of how Paschen curves are obtained for gases between electrodes (Supplementary Discussion 2 and Supplementary Fig. 3). Moreover, in pseudo-constant-pressure tests at low target pressures, minor discharge events are observed (before the first significant breakdown, labeled in Fig. 3a) at conditions relatively distant from predictions by Paschen's law, where successive drops of Coulomb force are detected at increasing voltages indicating that these do not represent intersections with the first half of the Paschen curve. Discharge in contact electrification at similar conditions on the left of the Paschen curve has been reported[7], as post-processed in a separate work[33]. This is hypothetically attributed to the increased significance of neutral particles being sputtered or evaporated[34–37] from the charged surfaces and participating in the avalanches of ionizations, since an increased gap distance at a pseudo-constant (reduced) electric field promotes the population of high-energy impinging electrons or cations, while the observation that each successive minor discharge event happens at a higher voltage may again be attributed to the increased diffusion of such neutral particles into the surroundings at larger gap distances.

The above results illustrate the characterization of gas breakdown in the contact electrification of insulators. The same test strategies are

applicable when one or both surfaces are conductive if the effect of induced surface charge redistribution is assumed minimal under the infinite-parallel-plate assumption. Given a surface with known secondary-electron-emission behaviors, the test setup can be used to estimate the ionization energy of an operating gas, while given a gas with known constituents it can be used to quantify the secondary electron emission from a test surface if it gains negative charge upon contact with the opposite. Here PDMS is selected as one of the surface materials for its high tendency to gain negative charge in contact, inferred by experimentally established triboelectric series[38,39], as well as its appropriate stiffness which is low enough to guarantee intimate contact with the other surface and thus uniform charge distribution while high enough for contacting surfaces to disengage within reasonable deformation so that gas breakdown is not triggered during the separation phase of contact electrification cycles. Moreover, the charging of PDMS is consistently efficient where fewer than 10 contact cycles against acrylic are sufficient to yield a significant surface charge density, while it has been observed that the charging of PDMS against PTFE (polytetrafluoroethylene), which has an even higher negative charge affinity, becomes less efficient after breakdown discharge at low gas pressures. This may be attributed to surface erosion caused either by mechanical compression and friction during contacts or more likely by sputtering under the cation bombardment in the gas breakdown process, which is a potential limitation in the application of the presented test strategies. Effective charging of harder surfaces using the same test setup may require extra contact force, finer surface topography as well as the introduction of friction[24], posing greater challenge on the overload protection of the load cell measuring Coulomb force, especially after aforementioned surface erosion occurs. Such extension of the experimental approach to surfaces with higher rigidity is illustrated in the following tests on nitrogen breakdown between a copper-nylon contact pair.

### Nitrogen discharge in copper-nylon electrification

The application of the proposed experimental method on gas breakdown between surfaces with inferior smoothness and compliance faces challenges in the efficiency of charge deposition under contact forces within the safety overload of the load cell used for measuring Coulomb forces. This is critical in situations where soft candidates such as PDMS cannot be used as one of the surfaces, e.g., when secondary electron emission from metal surfaces is to be characterized, since most of these surfaces gain positive charge in contact against PDMS. The following demonstrates preliminary strategies on enhancing charge transfer between non-elastomeric surfaces with test results of nitrogen discharge in copper-nylon electrification. Nylon (6-6) is selected as it gains positive charge in contact against copper (the polarity is both inferred by triboelectric series and verified experimentally), so that gas breakdown resulting from copper-nylon electrification is sustained by secondary electron emission from the copper surface. The nylon surface sample (film) is backed with a PDMS foundation (explained in Methods) to improve the effective contact area as well as to avoid plastic deformations during contacts. The contact area remains 45.6 cm² circular and the same pseudo-constant-pressure and -distance test procedures are followed to measure the breakdown voltage of nitrogen. Friction (rubbing) is introduced during the contact cycles (Fig. 4a), where the nylon sample is rotated 57.6° at peak compression force (20 N, 4.4 kPa) and rotated back after the surfaces are fully separated. Results of pseudo-constant-pressure and -distance tests and the collected breakdown voltage measurements are presented in Fig. 4b–d, respectively. It shows that with copper as the negatively charged surface (cathode), gas discharge on the left of the Paschen minimum is comparatively mild, i.e., the decrease of gap voltage at each breakdown event is small. Meanwhile, minor discharge events at low distance-pressure products prior to Paschen predictions are again observed. Energy dependence of the effective secondary-

electron-emission coefficient for copper surface in nitrogen breakdown is estimated and shown in Fig. 4e with comparison to that extracted from Paschen curves in literature[40]. Moreover, in copper-nylon contact electrification cycles the van der Waals adhesion is relatively trivial so that pseudo-constant-pressure tests may be initiated when the gap is closed.

### Supplementary measurements via applied voltage

The presented method aims to directly characterize gas breakdown triggered by triboelectric surface charge. For comparison, nitrogen breakdown is tested in a conventional setup (described in Fig. 5a and Methods) between metal electrodes with an externally applied voltage. Constant-distance tests are performed where breakdown events are recorded at various gas pressures. The breakdown voltage of nitrogen between aluminum electrodes at a gap distance of 1 mm, where the cathode is coated with PDMS, is depicted in Fig. 5c along with comparison to results from Fig. 3c, and that with copper cathode and aluminum anode without coating at a gap distance of 0.8 mm is shown in Fig. 5d along with results from Fig. 4d. Higher breakdown voltages in regions near the Paschen minimum are observed, which is attributed to factors discussed in nitrogen-PDMS-acrylic test results. Deviations may also be due to the current threshold (trip) used for breakdown detection in the high voltage power supply setup so that the Coulomb-force method is in general more sensitive to minor currents, while slight inconsistency in tests with PDMS-coated cathode may be related to charge accumulation on the dielectric surface altering electron states on the (coated) cathode surface and therefore affecting its secondary emission properties.

## Discussion

In the presented results, the selection of sample dimensions, load cell capacities as well as the target gas pressures and gap distances aims at proving the engineering feasibility of revealing complete Paschen curves including the minimum voltage point, i.e., to exhibit a gap voltage that survives all combinations of pressure and distance, as well as a clear connection between results from paths 1 and 2 in the pseudo-constant-distance tests, starting from low and high gas pressures respectively, without loss of data resolution. The amount of surface charge loss in each breakdown event is generally random, so that practically the survival gap voltage is always lower than the theoretical minimum and results from multiple test runs need to be combined to approximate its value asymptotically. Breakdown events at higher voltages in further regions of the Paschen curve can be obtained via pseudo-constant-distance tests with a larger target gap distance, or by increasing the number of contact cycles to deposit more surface charge. In this case, variations of $\gamma_{se}$ with respect to the surface charge density cannot be excluded since it is anticipated that secondary electron emission from an insulator surface with filled bands is more probable. Typically, the saturation level of surface charge density in a contact pair of materials distinct on the triboelectric series can generate a Coulomb force with an order of magnitude comparable to the contact force and the van der Waals adhesion, so that practically they can be measured using the same load cell. The presented test apparatus can therefore be used to quantify the buildup of surface charge by monitoring all surface interaction forces during contact cycles under low gas pressures with confidence that gas breakdown of Paschen (Townsend) type has not been triggered. As a brief demonstration, load cell readings in continuous contact cycles between PDMS and acrylic surfaces with a reduced effective contact area of 15.6 cm² (circular, 44.5 mm diameter) are shown in Fig. 6. In each cycle the Coulomb force at an infinitesimal gap immediately before the surfaces engage (state 2 in Fig. 6d, e) is used to calculate the real-time surface charge density under the infinite-parallel-plate assumption. A separate test on PDMS-iron contact electrification with a further reduced effective contact area of 4.62 cm² (circular, 24.3 mm

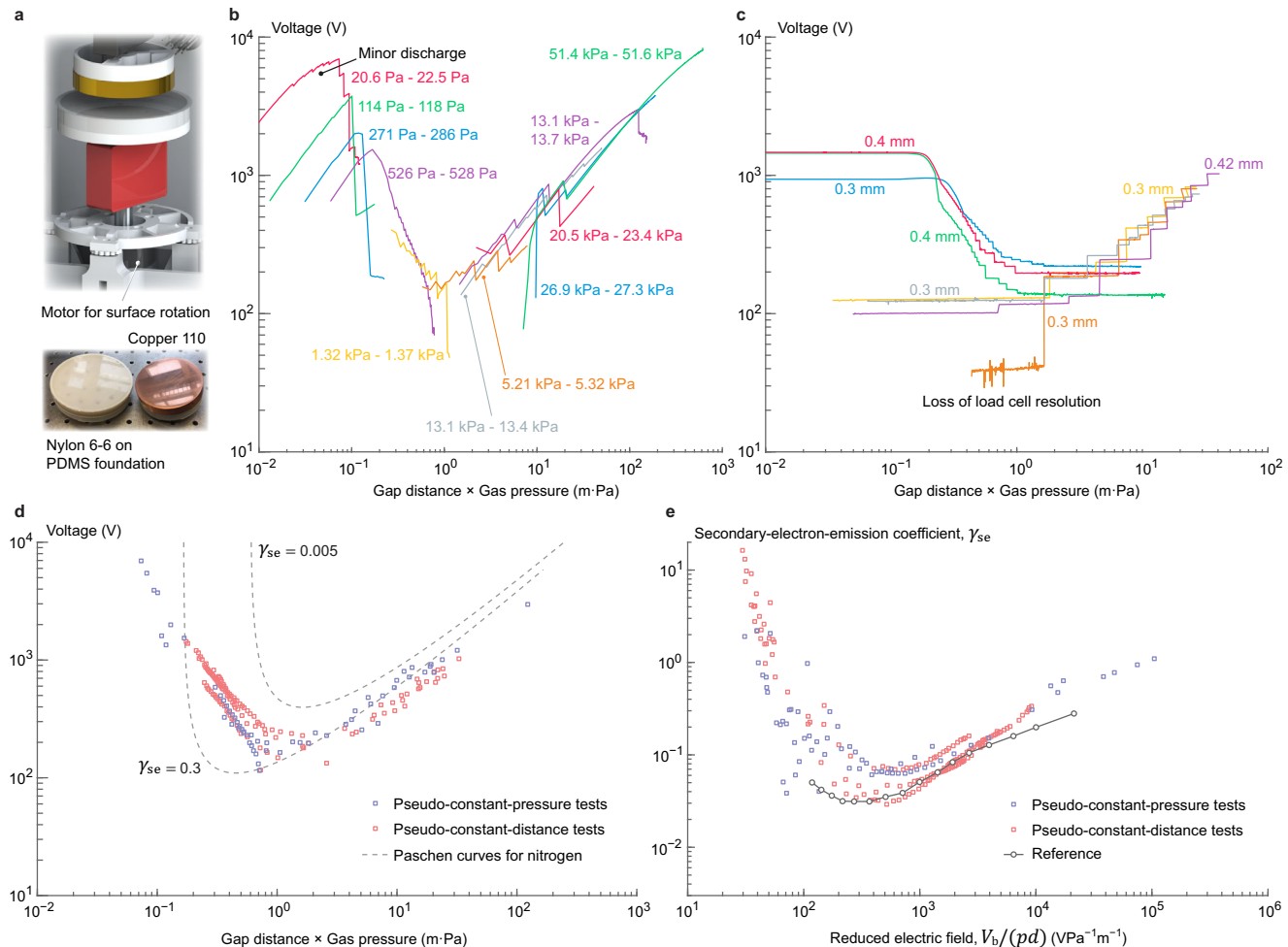

**Fig. 4 | Nitrogen breakdown in copper-nylon contact electrification. a** Adapted test apparatus to include surface friction driven by another stepper motor, where the top sample surface is copper and the bottom is nylon film backed with PDMS. **b** Pseudo-constant-pressure test results at multiple target gas pressures as labeled (in color) with tolerances. **c** Pseudo-constant-distance test results at multiple target gap distances labeled (in color) with their nominal values at zero load cell deflection. **d** Collected gap voltage readings at breakdown events detected from test runs in both strategies (lavender are pseudo-constant-pressure tests results, coral pseudo-constant-distance). Dashed lines are reference curves evaluated using the classical Paschen's law (1) with empirical coefficients $A$ and $B$ for nitrogen while assuming two hypothetical values of $\gamma_{se}$. **e** Effective secondary-electron-emission coefficient of copper in nitrogen discharge, estimated from test results in (**d**) by plugging gap voltage, gas pressure and gap distance at each measured breakdown event into classical Paschen's law (1), with comparison to reference values (medium gray) calculated in the same way using breakdown voltage data extracted from literature[40].

---

diameter) indicates a nearly saturated surface charge density of approximately 480 μC/m² (Supplementary Discussion 3 and Supplementary Fig. 4), which is comparable in magnitude to values reported in literature[41]. This may facilitate investigations of charge transfer mechanisms in contact electrification of different materials based on either the saturation level of charge density or the trend of accumulative charge deposition by repeated contacts.

## Methods

### Sample fabrication and test setup

In tests of nitrogen breakdown in PDMS-acrylic contact electrification the bottom sample (acrylic disc) is 76.2 mm (3 inches) in diameter and 6.4 mm (1/4 inches) thick (commercially available, McMaster-Carr). The top sample (PDMS) is fabricated following steps shown in Fig. 1c with details displayed in Supplementary Fig. 5a. The 3D-printed mold (with ASA filament, on Prusa MK3S) is pressed against a clear polyester sheet (Grafix, 0.18 mm thickness, cleaned with isopropyl alcohol) on a flat surface, and then degassed liquid PDMS (Sylgard 184) is poured inside via an array of channels on its bottom. The PDMS is then cured in

atmosphere under room temperature for 48 h, while the channels ventilate extra bubbles generated during the casting and curing processes. Overhang structures printed on the floor of the mold serve as buried mechanical locks that seize the cured PDMS to prevent it from peeling off the mold floor or walls under strong contact or van del Waals forces (Fig. 6) in contact electrification cycles. The overhangs are 3.4 mm in height and the bulk PDMS above is 6.6 mm thick. In tests of nitrogen breakdown in copper-nylon contact electrification the top sample (copper disc) is 76.2 mm (3 inches) in diameter and 12.7 mm (1/2 inches) thick (copper 110, commercially available, McMaster-Carr), sanded up to 7000 grits. The bottom sample (nylon) is fabricated by pre-stretching and pressing a nylon film (nylon 6-6, 25 μm thickness, commercially available, McMaster-Carr) on a glass disc and then casting PDMS (Ecoflex 00-30, Smooth-On) in a mold fixed on the nylon film along its perimeter to form an elastomeric foundation. The PDMS is degassed both before and after casting to eliminate residual gas trapped in the mold as well as between the PDMS and the nylon surface.

The test setup is illustrated in Fig. 1b with details explained in Supplementary Discussion 4 and Supplementary Fig. 5. The acrylic

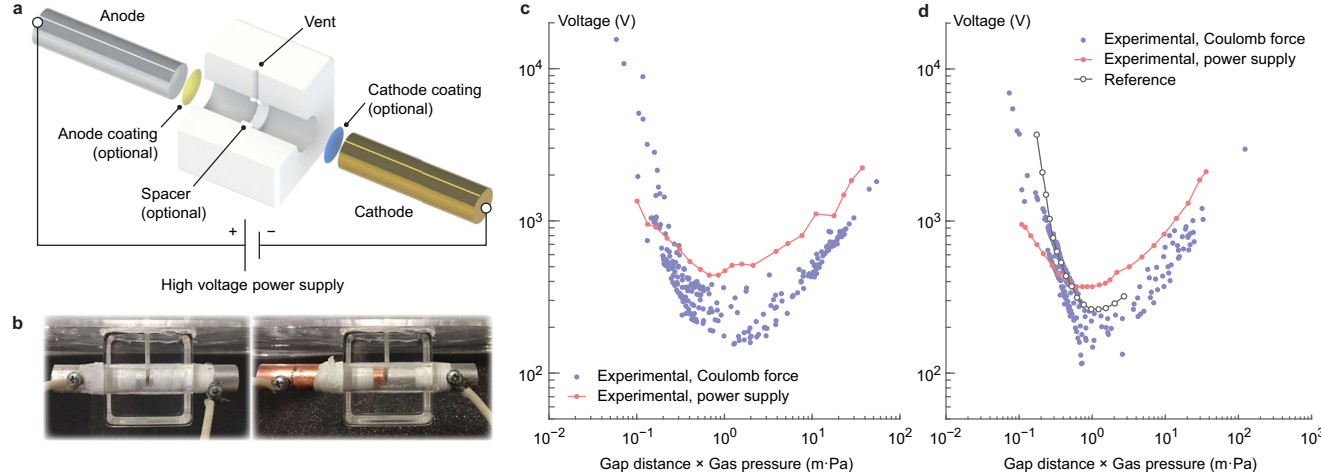

**Fig. 5 | Supplementary tests using a high voltage power supply for validation of test results on nitrogen discharge. a** Schematics of test setup. **b** Pictures of (Left) PDMS(coated cathode)-aluminum(anode) and (right) copper(cathode)-aluminum(anode) setup. **c** Measured breakdown voltage of nitrogen with PDMS-coated cathode (coral using power-supply setup, lavender from test results via Coulomb force in Fig. 3c). **d** Measured breakdown voltage of nitrogen with copper cathode (coral using power-supply setup, lavender from test results via Coulomb force in Fig. 4d), with comparison to reference data (medium gray) extracted from literature[40].

vacuum chamber is customized (Sanatron) to be 305 mm (12 inches) cubic with inlet and outlet connected to the gas cylinder (Indiana Oxygen, nitrogen, >99.998% purity) and a 2-stage mechanical pump (Across International SuperVac-5C, 5.6 cfm, mounted on a separate table for vibration isolation), respectively. Gas pressure in the chamber is measured via a Pirani gauge (Instrutech Stinger CVM 211) with a log-linear output calibrated for nitrogen. The top sample is mounted on the high-capacity load cell (Mark-10, MR03-5, 25 N capacity), whose vertical motion is driven by 2 synchronized stepper motors with a displacement resolution of 0.02 mm, where motor heat is dissipated by conduction through the chamber body. The bottom sample is fixed on the low-capacity load cell (Futek, LRF 400, 1.2 N capacity) which is mounted on the base by 4 leveling screws. The overload of the bottom load cell during the contact cycles is kept within its safety level of 200 N. The surface alignment error is calibrated to under 0.02 mm using the leveling screws by sliding a thin strip of paper from several directions into the gap and then comparing the friction when the paper is being pulled out after the gap is closed (Supplementary Fig. 5c). Drift and hysteresis of the bottom load cell readings under gas pressure variations are shown in Supplementary Fig. 5e where the load cell is zeroed when the surfaces are separated to a gap of 30 mm and the gas pressure is swept from 20 Pa to 100 kPa for 2 cycles. This error is not compensated for in the results reported in Figs. 3 and 4 since in pseudo-constant-distance tests the deviation is trivial at high voltages while at low voltages the gas pressure is in general lower than 10 kPa where the load cell error is within 0.1 mN, and in pseudo-constant-pressure tests the breakdown events under pressures higher than 10 kPa generally occur at small gap distances where load cell error (<1 mN) is trivial compared to the magnitude of the Coulomb forces corresponding to the breakdown voltages. Readings of 3 typical contact cycles in the charging phase of a test run are shown in Supplementary Fig. 6 for the bottom load cell, from which a linear stiffness of 4.57 N/mm (Supplementary Fig. 6c) is estimated for deflections of the load cells to be compensated for in the recorded gap distance and the calculation of gap voltage afterwards. In tests requiring friction (rubbing) between the sample surfaces, the bottom load cell is instead mounted on another stepper motor fixed on base. Differences in the observed van del Waals adhesion between PDMS-acrylic and copper-nylon tests are presented in Supplementary Discussion 5 and Supplementary Fig. 7. Customized structural parts in the test apparatus are 3D-printed in ASA and PLA, CAD files available upon request, and all samples are kept away from any grounded conductor to prevent any induced

charge from disturbing the electric field. The load cell output and stepper motor input are transmitted via a wire feedthrough on the back of the vacuum chamber, for which disturbance and noise are minimized by disabling the stepper motors 0.4 s ahead whenever a Coulomb force reading is taken. Data acquisition, visualization and test programming are integrated in a user interface on the Qt framework with its serial communication module. Time traces of Coulomb force measurements in representative test runs from Fig. 3 are presented in Supplementary Discussion 6 and Supplementary Fig. 8.

**Evaluation of gap voltage by Coulomb force**

In the demonstrated tests of nitrogen breakdown between charged surfaces, assuming a uniform surface charge density $\pm\sigma$ on both samples (circular), the magnitude of the attractive Coulomb force at gap distance $d$ is given by

$$F_c = \frac{\sigma^2 d}{2\varepsilon_{gas}} \int_0^{2\pi} \int_0^R \int_0^R \frac{r_1 r_2}{\left(r_1^2 + r_2^2 - 2r_1 r_2 \cos\theta + d^2\right)^{3/2}} dr_1 dr_2 d\theta \quad (2)$$

where $\varepsilon_{gas} \approx \varepsilon_{vacuum} \approx 8.85 \times 10^{-12}$ F·m$^{-1}$ is the permittivity of the operating gas and $R$ the sample radius. Once the surface charge density is derived from the Coulomb force reading, the corresponding gap voltage is evaluated using the infinite-parallel-plate assumption so that $V = \sigma d/\varepsilon_{gas}$. This is based on the assumption that gas breakdown events are generally triggered at locations with the highest voltage, in this case the center of the disks, where the voltage across the gap along a linear path connecting the center of the two surfaces is

$$V = \frac{\sigma}{4\pi\varepsilon_{gas}} \int_0^d \int_0^{2\pi} \int_0^R \frac{hr}{\left(r^2 + h^2\right)^{3/2}} dr d\theta dh \quad (3)$$

so that, for example, in the presented test runs the error of voltage estimation is kept below 5% when the gap distance is lower than 4 mm.

**High voltage power supply setup**

In comparison tests utilizing (coated) metal electrodes with applied voltage, a customized chamber is constructed in a 38.1 mm (1.5 inch) cubic acrylic block (Fig. 5a, b). The electrodes are cylindrical, 12.7 mm (1/2 inches) in diameter, and separated at a fixed gap in a drilled

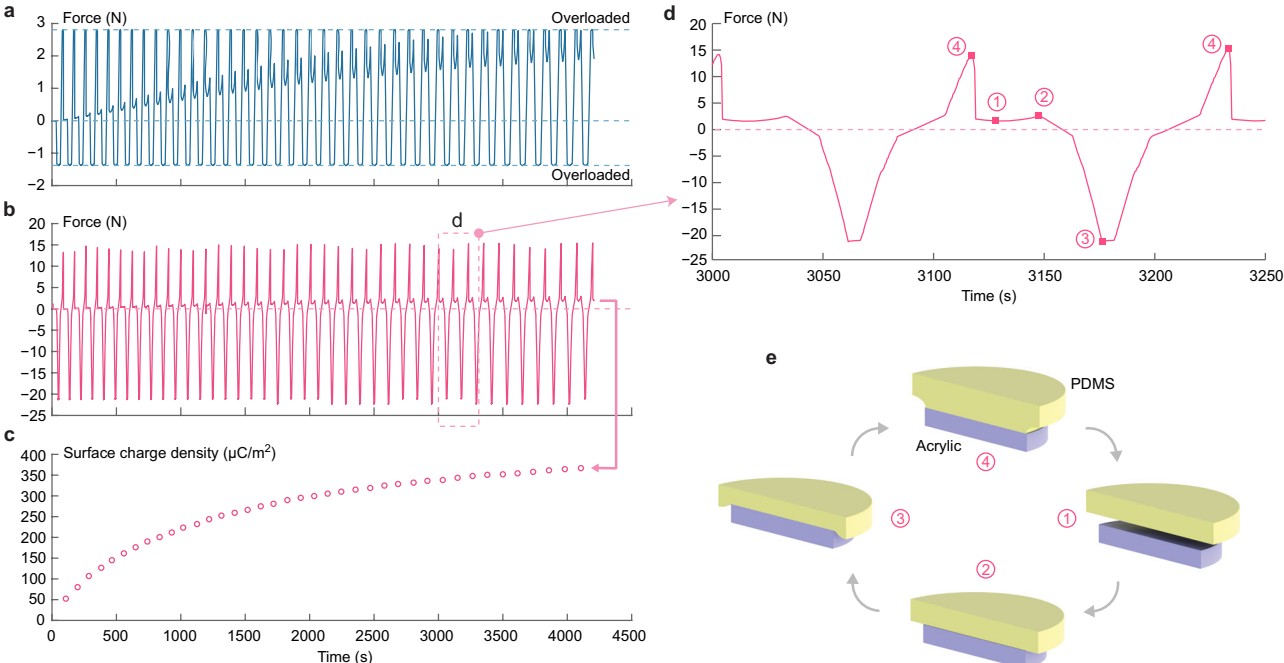

**Fig. 6 | Surface charge accumulation in PDMS-acrylic contact electrification cycles. a** Time history of the bottom load cell (low capacity) readings, positive being attraction. **b** Time history of the top load cell (high capacity) readings, positive being attraction. **c** Surface charge density estimated via Coulomb force readings at time instants of infinitesimal gap distance immediately before the surfaces engage in each cycle. **d** Details of the top load cell reading in a labeled time window, with cyclic states demonstrated in (**e**) including (1) surfaces separated to maximum distance, (2) surfaces at an infinitesimal gap immediately before engaging, (3) surfaces compressed to peak contact force and (4) surfaces experiencing peak van der Waals adhesion immediately before disengaging.

channel. The chamber is connected to the vacuum pump and gas inlet via a vent, while outside the chamber the electrodes are directly wired to a high voltage power supply (Stanford Research Systems PS375). Breakdown voltage is measured by maintaining a constant target gas pressure and then increasing the applied voltage with a step of 10 V until a current exceeding 500 μA triggers a trip in the power supply. PDMS coating is applied by gravity casting on the aluminum electrode to form a layer of approximately 20 μm thickness, during which it is ensured to cover all edges and sides of the electrode to eliminate alternative discharge channels formed by exposed metal surfaces. All electrodes are sanded (before coating), cleaned with isopropyl alcohol and baked at 120 °C for 1 h (after coating) before testing.

## Data availability

The gas discharge data generated in this study have been deposited in the Figshare database under accession code https://doi.org/10.6084/m9.figshare.23269808.

## Code availability

Source code for the user interface of the customized test apparatus is available at https://github.com/adamsPurdue/pasc.

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

## Acknowledgements

The authors thank Professors Chelsea Davis (University of Delaware), Arvind Raman (Purdue University) and Jeffrey Rhoads (University of Notre Dame) for discussions and advice. The authors thank the faculty and staff members of the Ray W. Herrick Laboratories for facilitating the experimental setup. This work is supported by the National Science Foundation under grant CMMI 1662925 (to J.G.) and CAREER Award CMMI 2145803 (to J.G.). Publication of this article was funded in part by Purdue University Libraries Open Access Publishing Fund.

## Author contributions

J.G. led the proposal of the methodology and analysis. H.T. imple-mented the experimental setup. H.T. and J.G. processed the data and prepared the manuscript.

## Competing interests

The authors declare no competing interests.
