## [Peer Review File · Nature Communications]

REVIEWER COMMENTS

Reviewer #1 (Remarks to the Author):

The paper proposes a nondestructive method similar to that reported by other authors in a prior work which uses Coulomb force measurements to monitor surface charge variations and thus quantify the breakdown voltage of the gas between electrified surfaces with respect to its pressure and the gap distance.

The paper really proposes a good method and provides the experimental evidence for a Paschen-type behavior of nitrogen breakdown between a silicone-acrylic pair. However, the paper is not recommended for publication in Nature communications for the following two reasons. First, the similar method had been reported by other authors nearly 30 years ago. Second, such a breakdown obeying Paschen-law is predictable.

I would like to suggest the paper be submitted to Review of Scientific Instruments. Before the submission, it is better to validate the results of the measurement. For example, you may use two plane-parallel metal electrodes with the cathode covered with the silicone film or the acrylic film. Then, you can measure and compare the breakdown voltages under the same experimental conditions.

Reviewer #2 (Remarks to the Author):

This work presents an investigation on a method to measure gas discharges between insulating bodies charged through contact electrification.

The authors use simultaneous measurements of separation and electrostatic force between the separating bodies at the moment of the discharge to calculate the voltage between them assuming a parallel plate geometry at different gas pressures. The results and analysis are clearly stated, and experiments seem to have been made carefully and methodically.

1. A very similar methodology was employed in the following work, that I think should be cited:

Collins, Adam L., et al. "Simultaneous measurement of triboelectrification and triboluminescence of crystalline materials." *Review of Scientific Instruments* 89.1 (2018).

2. When compared to both this reference and Horn's 1992 paper (ref. 7), the contribution from the present work should be made absolutely clear.

I strongly recommend this because, at first glance, it seems that the authors are able to reconstruct Paschen's law which would be enough proof of the validity of their results and the novelty of their research. However, as the authors state themselves, they need to assume at least two different values of the secondary electron emission coefficient to reconcile their results with Paschen's law. Furthermore, as they show in Fig. 4, a coefficient for these materials varying by more than four orders of magnitude would be needed. But the authors do not provide independent measurements of this coefficient as a function of electron energy (other than some references). Consequently, it seems that they use the secondary coefficient as a fitting parameter, which would make their results less interesting, specially compared to the works cited above. In this respect, saying that they obtain a "Paschen-type behavior" as they do in the abstract, it really not that surprising.

3. It would be ideal to see in the paper the same experiments between conductors (whose secondary coefficients are better known) to reassure the reader that they experimental setup gives the expected results with conventional materials (like, say, copper). But I understand that in that case an external high-voltage supply would be necessary, along with a fast switch which may not be straightforward to implement.

4. Alternatively, I suggest that the authors should include at a plot of the secondary emission coefficient for these materials vs. electron energy (either from their own research or values found in the literature) to show that the values they infer with their results are indeed consistent if not similar to those measured independently. Perhaps add a discussion about the energy range in which they match or not. Presently, in this respect they do mention trends, but the plot I suggest would make it more evident for the reader that the secondary emission coefficient is not simply used as a fitting parameter.

5) The authors should include a discussion about how the magnitude of the surface charge they infer in the supplemental material compares to that found in the literature.

I recommend the publication of this paper as long as the changes proposed are carried out by the authors.

Reviewer #3 (Remarks to the Author):

This manuscript should be accepted for publication in Nature Communications. The paper showed direct evidence for Paschen based gas discharge between two polymer plates. The results must be welcomed by the community of triboelectrification.

Just for readability of the document, some minor revision may be suggested:

1. Figure 2 looks a little too busy or noisy. The drawings of electrodes may not necessary.
2. Presenting some examples of recorded signals of force measurement with gas breakdown events may be helpful to understand the experiments.
3. As discussed in figure 5, there might be a 'jump-off' at the disengaging after Van der Waals adhesion. Please describe the gap distance after the disengaging.
4. Figure 4 could show the recognition between the results of the pseudo-constant-distance tests and the pseudo-constant-pressure tests.

Measuring gas discharge in contact electrification

Hongcheng Tao¹, James Gibert¹

¹School of Mechanical Engineering, Purdue University, West Lafayette, IN, USA.

Summary

We (the authors) would like to thank the editor and reviewers for dedicating their valuable time and expertise to assess the manuscript. We appreciate the reviewers' thorough and insightful comments, which significantly contributed to enhancing the quality of the present work. In the following we elucidate our response to each remark, along with the corresponding revisions made in the manuscript, in our pursuit of continued consideration for this submission.

Major changes include:

- A new subsection describing new test results for nitrogen breakdown in copper-nylon electrification
- A new subsection on supplementary test results for gas breakdown using a high voltage power supply
- Supplementary results for charge accumulation in Fe-PDMS electrification
- A supplementary video of load cell deflections and oscillations

The major edits are highlighted in **blue** in a separate copy of the revised manuscript, as well as attached to related reviewer remarks in this response letter for convenience of reference. Additionally, the structure of the main text is briefly adjusted according to formatting instructions of resubmission to *Nature Communications*.

Response to Reviewer #1

General remark

The paper proposes a nondestructive method similar to that reported by other authors in a prior work which uses Coulomb force measurements to monitor surface charge variations and thus quantify the breakdown voltage of the gas between electrified surfaces with respect to its pressure and the gap distance.

The paper really proposes a good method and provides the experimental evidence for a Paschen-type behavior of nitrogen breakdown between a silicone-acrylic pair.

Response

We appreciate the reviewer's evaluation of the present work.

Remark - 1

However, the paper is not recommended for publication in Nature communications for the following two reasons.

First, the similar method had been reported by other authors nearly 30 years ago.

Response

We appreciate the reviewer's evaluation of the experimental approach, and we agree that the method of measuring surface charge density in contact electrification via Coulomb force has been proposed and adopted readily in literature as the physics is clearly fundamental. We would like to clarify that in the original submission we did not intend to claim novelty of the present work on the successful implementation of surface charge density measurement (despite minor contributions in technical details). Instead, we hope we may have the opportunity to explain to our reviewer on the present work's more important contributions both in *applying the Coulomb-force method on the characterization of gas breakdown* and in the *examination/validation of the assumed Paschen curves governing gas breakdown in contact electrification* (discussed in the response to Remark 2). The apparent challenge of achieving these goals lies in acquiring non-trivial measurements of breakdown

voltage near the minimum point on Paschen curves. While there is typically already a significant difference in the magnitudes of the Coulomb force and the contact force required to deposit the surface charge, the Coulomb force corresponding to the lowest breakdown voltages can be even orders of magnitude smaller. For example, in the presented tests of nitrogen breakdown in PDMS-acrylic electrification, the peak contact force during the charging cycles is 20 N, the pre-breakdown Coulomb forces are between 0.1 N to 1N, while the Coulomb forces measured at breakdown events near the Paschen minimum are typically around 0.01N. In particular, we define a major objective of the present work as to show clear evidence that a complete Paschen curve can be obtained with the proposed setup and we demonstrated such ‘completeness’ by showing the connectivity among multiple segments of the Paschen curve recorded in tests under various conditions, as well as by showing a non-trivial residual (survival) surface charge density after the gas pressure is swept over the entire range of interest in pseudo-constant-distance tests. While technically straightforward, we learned that such strategies of reconstructing Paschen curves, as shown in Fig. 2 of the original manuscript, are not currently derivable from literature, and we thus believe that the experimental proof of the engineering feasibility of acquiring complete Paschen curves using the Coulomb-force setup is a novel and fundamental contribution potentially helpful for future related works to refer to.

Remark - 2

Second, such a breakdown obeying Paschen-law is predictable.

Response

We appreciate the comment and we agree that the classical Paschen’s law is generally applicable as a guideline in the modeling and analysis of gas breakdown in contact electrification. It is indeed the fact that Paschen’s law has been assumed so widely in studies concerning gas breakdown triggered by surface charge that motivated the present work to provide an experimental validation for this assumption, for which we learned that insufficient evidence is available from literature so far. As we believe in principle that all theories may benefit from direct experimental examination, we hope we may have the opportunity to explain to our reviewer on the present work’s contributions in and beyond the reconstruction of Paschen curves in contact electrification. As a theory that continues to develop, the formulas of the classical Paschen’s law as well as its various amended versions consist of parameters dependent on properties of both the operating gas and the surface materials. These parameters as well as potential corrections to the conventional theory need to be experimentally determined for each independent scenario, where numerous ongoing efforts are being made in literature to characterize the discharge behavior of different gases between different surfaces (electrodes), despite that in most cases the obedience of classical Paschen’s law is qualitatively predictable. In fact, our test results readily exhibited deviations from predictions of the classical Paschen’s law which is mainly attributed to the energy dependence of the secondary-electron-emission coefficient originally assumed constant in the formula. We were able to extract this information from the measured Paschen curve, as depicted in Fig. 4 of the original manuscript, which matches both in trend and magnitude with reported results from literature on various gases and electrode materials. It is for this same reason that while the rudimentary goal of the present work was to ‘verify Paschen’s law in contact electrification’, we decided to be cautious and precise on the statement and instead described our findings as a Paschen-type behavior (rephrased using ‘Paschen curves’ in the revised manuscript). Based on these results, we attempted to discuss the broader applications of the present work in serving as a preliminary investigation of alternative methods to characterize gas breakdown in general. (*Casually speaking, we would like to explore the possibility of proceeding from ‘measuring gas discharge in contact electrification’ to ‘measuring gas discharge using contact electrification’.*) Despite practical challenges including quality and efficiency of charging between arbitrary surfaces, the presented method may arguably possess potential advantages compared to the conventional setup with a high voltage power supply. For example, in the conventional setup the geometries of the electrodes and the vacuum chamber may require specific designs to avoid undesired discharge paths, e.g., when a gap is formed between two exposed points that is either shorter or longer than the designated one so that it triggers early gas breakdown, while in the Coulomb-force setup with the absence of any circuit or undesired conductive parts the electric field is comparatively localized in the designated clearance gap. Meanwhile, in conventional tests using insulator/dielectric coatings on electrodes, the charge accumulation on the dielectric surfaces resulting from each breakdown event is difficult to either quantify or nullify even with an AC setup, which may affect electron states and thus secondary-emission properties of the surface or potentially break down the dielectric layer itself, while in the Coulomb-force setup since the polarity comes directly from transferred surface charge it is guaranteed that the dielectric surfaces are being neutralized during each breakdown event. We sincerely hope that the above may reassure our reviewer of the necessity and potential applications of the experimental results in the present work.

Remark - 3

I would like to suggest the paper be submitted to Review of Scientific Instruments. Before the submission, it is better to validate the results of the measurement. For example, you may use two plane-parallel metal electrodes with the cathode covered with

the silicone film or the acrylic film. Then, you can measure and compare the breakdown voltages under the same experimental conditions.

Response

We thank the reviewer for the recommended submission to the prestigious journal *Review of Scientific Instruments*. With thorough deliberations, we regret to say we have concluded that the present work may not align with its intended scope optimally. As clarified in our response to Remark 1, we do not claim the current work's contributions on the implementation of an instrument that performs controlled contact electrification and measures surface charge density via Coulomb force. Despite a few novel technical features, we are concerned that excessive emphasis on the design of our experimental apparatus may inevitably overshadow the more significant aspects of the present work. That is, in short, we would like to focus on the proposed methodology of measuring ubiquitous surface-charge-induced gas breakdown, and we believe that the presented experimental evidence of complete Paschen curves reconstructed free of circuits represents a non-trivial addition to the literature. We also believe that interpretations and applications of these results are potentially comprehensive for being useful in various subjects where the characterization and modeling of gas breakdown are concerned, as readily discussed in the manuscript. Therefore, we are honored to take the opportunity to resubmit our revised work to *Nature Communications* in the hope of broadening the exposure of this work to readers from interdisciplinary backgrounds.

We appreciate very much the suggestion of supplementary tests for validation. In the revised manuscript we included results and discussion of breakdown tests on nitrogen with PDMS-coated cathode in a conventional setup utilizing a high voltage power supply. These results yielded a good match with the Coulomb-force approach while sources of deviations and inconsistencies are discussed in the manuscript as well as attached below. Attached below before that is a new set of tests that we appended on nitrogen breakdown in copper-nylon electrification, which we would also like to bring to our reviewer's attention if possible. This is to address the method's applicability on different surface materials so that secondary-electron-emission properties of any material of interest may be investigated by proper choice of the opposite surface and thus the charging polarity. Comparison tests with a power supply are also conducted for this configuration and since copper serves as the cathode we were able to extract reference from literature to provide further validation of our test results.

Revisions

In Results:

Nitrogen discharge in copper-nylon electrification

The application of the proposed experimental method on gas breakdown between surfaces with inferior smoothness and compliance faces challenges in the efficiency of charge deposition under contact forces within the safety overload of the load cell for measuring Coulomb forces. This is critical in situations where soft candidates such as PDMS cannot be used as one of the surfaces, e.g., when secondary electron emission from metal surfaces is to be characterized, since most of these surfaces gain

(New) Fig. 5 | Nitrogen breakdown in copper-nylon contact electrification. **a** Adapted test apparatus to include surface friction. **b** Pseudo-constant-pressure test results at multiple target gas pressures as labeled with tolerances. **c** Pseudo-constant-distance test results at multiple target gap distances labeled with their nominal values at zero load cell deflection. **d** Collected gap voltage readings at breakdown events detected from test runs in both strategies.

positive charge in contact against PDMS. The following demonstrates preliminary strategies on enhancing charge transfer between non-elastomeric surfaces with test results of nitrogen discharge in copper-nylon electrification. Nylon (6-6) is selected as it gains positive charge in contact against copper (the polarity is both inferred by triboelectric series and verified experimentally), so that gas breakdown resulting from copper-nylon electrification is sustained by secondary electron emission from the copper surface. The nylon surface sample (film) is backed with a PDMS foundation (explained in Methods) to improve the effective contact area as well as to avoid plastic deformations during contacts. The contact area remains 45.6 cm² circular and the same pseudo-constant-pressure and -distance test procedures are followed to measure the breakdown voltage of nitrogen. Friction (rubbing) is introduced during the contact cycles (Fig. 5a), where the nylon sample is rotated 57.6° at peak compression force (20 N, 4.4 kPa) and rotated back after the surfaces are fully separated. Results of pseudo-constant-pressure and -distance tests and the collected breakdown voltage measurements are presented in Figs. 5b, 5c and 5d, respectively. It shows that with copper as the negatively charged surface (cathode), gas discharge on the left of the Paschen minimum is comparatively mild, i.e., the decrease of gap voltage at each breakdown event is small. Meanwhile, minor discharge events at low distance-pressure products prior to Paschen predictions are again observed. Energy dependence of the effective secondary-electron-emission coefficient for copper surface in nitrogen breakdown is estimated and shown in Fig. 6 with comparison to that extracted from Paschen curves in literature⁴⁰. Moreover, in copper-nylon contact electrification cycles the van der Waals adhesion is relatively trivial so that pseudo-constant-pressure tests may be initiated when the gap is closed.

(New) Fig. 7 | Supplementary test with applied voltage. **a** Schematics of test setup. **b** (Left) PDMS(coated cathode)-aluminum(anode) and (right) copper(cathode)-aluminum(anode) setup. **c** Measured breakdown voltage of nitrogen with PDMS-coated cathode. **d** Measured breakdown voltage of nitrogen with copper cathode, with comparison to literature⁴⁰.

Supplementary measurements via applied voltage

The presented method aims to directly characterize gas breakdown triggered by triboelectric surface charge. For comparison, nitrogen breakdown is tested in a conventional setup (described in Fig. 7a and Methods) between metal electrodes with an externally applied voltage. Constant-distance tests are performed where breakdown events are recorded at various gas pressures. The breakdown voltage of nitrogen between aluminum electrodes at a gap distance of 1 mm, where the cathode is coated with PDMS, is depicted in Fig. 7c along with comparison to results from Fig. 3c, and that with copper cathode and aluminum anode without coating at a gap distance of 0.8 mm is shown in Fig. 7d along with results from Fig. 5d. Higher breakdown voltages in regions near the Paschen minimum are observed, which is attributed to factors discussed in nitrogen-PDMS-acrylic test results. Deviations may also be due to the current threshold (trip) used for breakdown detection in the high voltage power supply setup so that the Coulomb-force method is in general more sensitive to minor currents, while slight inconsistency in tests with PDMS-coated cathode may be related to charge accumulation on the dielectric surface altering electron states on the (coated) cathode surface and therefore affecting its secondary emission properties.

We hope that in the above we have explained and clarified the methodology, results and contributions of the present work in a better way, and we hope that our reviewer may find our response to the remarks appropriate.

Response to Reviewer #2

General remark

This work presents an investigation on a method to measure gas discharges between insulating bodies charged through contact electrification.

The authors use simultaneous measurements of separation and electrostatic force between the separating bodies at the moment of the discharge to calculate the voltage between them assuming a parallel plate geometry at different gas pressures. The results and analysis are clearly stated, and experiments seem to have been made carefully and methodically.

Response

We appreciate the reviewer's evaluation of the present work.

Remark - 1

1. A very similar methodology was employed in the following work, that I think should be cited:

Collins, Adam L., et al. "Simultaneous measurement of triboelectrification and triboluminescence of crystalline materials." *Review of Scientific Instruments* 89.1 (2018).

Response

We thank the reviewer for the recommended citation. The contributions of the given work and its direct relevance to the present work were recognized upon initial review, but our apologies that its inclusion in the original manuscript was inadvertently omitted. It is cited in the revised manuscript in related statements.

Revisions

In *Introduction*: ... The present work therefore implements an alternative nondestructive approach similar to setups reported in prior works^{7,24} which uses Coulomb force measurements to...

In *Results*: ... Meanwhile, effective charging of harder surfaces using the same test setup may require extra contact force, finer surface topography as well as the introduction of friction²⁴ ...

Remark - 2

2. When compared to both this reference and Horn's 1992 paper (ref. 7), the contribution from the present work should be made absolutely clear:

I strongly recommend this because, at first glance, it seems that the authors are able to reconstruct Paschen's law which would be enough proof of the validity of their results and the novelty of their research. However, as the authors state themselves, they need to assume at least two different values of the secondary electron emission coefficient to reconcile their results with Paschen's law. Furthermore, as they show in Fig. 4, a coefficient for these materials varying by more than four orders of magnitude would be needed. But the authors do not provide independent measurements of this coefficient as a function of electron energy (other than some references). Consequently, it seems that they use the secondary coefficient as a fitting parameter, which would make their results less interesting, specially compared to the works cited above. In this respect, saying that they obtain a "Paschen-type behavior" as they do in the abstract, it really not that surprising.

Response

We appreciate the comment and we agree that contributions of the present work may need further clarification. In particular, we appreciate the reviewer's identification of the current study's novelty in the reconstruction of Paschen curves for contact electrification, and we hope we can have the opportunity to explain our reasoning behind how we presented our work in the original manuscript. We have been aware that a statement that we 'experimentally validated Paschen's law for gas breakdown in contact electrification' would be ideal to conclude the findings, as it was the rudimentary motivation of the present work. However, we took extra caution to be precise about the wording since it has been evident, both in literature and our study, that the classical Paschen's law is typically a general approximation which may require corrections according to various scenarios. A major source of error is indeed the discussed energy-dependent secondary-electron-emission coefficient that is conventionally

assumed constant in Paschen's law. As a consequence, we decided to describe the experimental results as a Paschen-type behavior; which represents a general valley-shaped curve of breakdown voltage vs pressure-gap product. However, we agree with the reviewer that such statements may be vague, and therefore in the revised manuscript we rephrased 'Paschen-type behavior' using 'Paschen curves', which is more recognized while allowing corrections to the classical form. It is for this same reason that we did not attempt to use the entire set of recorded data to fit all parameters in the formula of Paschen's law. Instead, the effective secondary-electron-emission coefficients in Fig. 4 were calculated for each individual data point using this formula assuming constant parameters A and B for the gas. This step has been a common practice in literature for characterizing surface secondary electron emission using Paschen curves examined via various approaches¹⁻³. It is apparent that the energy dependence of the secondary-electron-emission coefficient shown in these works, although measured with different operating gases and cathode materials, is comparable to our results (Fig. 4) in both trend and magnitude. Based on these references we have not been concerned about its value varying by more than four orders of magnitude, as we are confident that such variation does not indicate errors in the tests but is instead a valid estimation of its true energy dependence. Moreover, we believe that this large span of magnitudes of the calculated coefficient may actually reveal the potential advantage of the present approach that it is capable of characterizing secondary electron emission for a comparatively large span of incident particle energy (e.g., compared to individual tests presented in references listed below), i.e., in this case nearly five orders of magnitude as shown on the horizontal axis of Fig. 4. (*Casually speaking, we obtained more than four orders of magnitude of the coefficient partly because we covered nearly five orders of magnitude of incident particle energy.*) We shall further illustrate this point in a more quantitative way in our response to Remarks 3 and 4 below.

References

1. Druyvesteyn, M. J., and Fi M. Penning. "The mechanism of electrical discharges in gases of low pressure." *Reviews of Modern Physics* 12.2 (1940): 87.
2. Auday, G., et al. "Experimental study of the effective secondary emission coefficient for rare gases and copper electrodes." *Journal of applied physics* 83.11 (1998): 5917-5921.
3. Phelps, A. V., and Z. Lj Petrovic. "Cold-cathode discharges and breakdown in argon: surface and gas phase production of secondary electrons." *Plasma Sources Science and Technology* 8.3 (1999): R21.

Related results

Fig. 19, Fig. 20
 Fig. 7, Fig. 8, Fig. 9, Fig. 10
 Fig. 5

Revisions

In Abstract: The present work implements ..., providing experimental evidence of Paschen curves governing nitrogen breakdown in silicone-acrylic and copper-nylon contact electrification.

In Introduction: The present work therefore implements an alternative nondestructive approach similar to setups reported in prior works^{7,24} which ... It is illustrated with the reconstruction of complete Paschen curves for nitrogen breakdown in both silicone-acrylic and copper-nylon contact electrification.

Remark - 3

3. It would be ideal to see in the paper the same experiments between conductors (whose secondary coefficients are better known) to reassure the reader that they experimental setup gives the expected results with conventional materials (like, say, copper). But I understand that in that case an external high-voltage supply would be necessary, along with a fast switch which may not be straightforward to implement.

Response

We appreciate this suggestion and we agree that test results with more sets of materials will definitely improve the completeness of the present work. In the original tests we used the PDMS-acrylic combination mainly to demonstrate the experimental approach's capability to measure gas breakdown between exclusively insulator/dielectric surfaces, in which the mechanical properties of PDMS have greatly eased the charging process. Yet a limitation lies in that PDMS gains negative charge against most materials except few (e.g., PTFE), so that it may not be used when secondary electron emission from the opposite surface (e.g., copper, as suggested) is of interest. Therefore, we conducted a new test for nitrogen breakdown with a copper-nylon contact pair where nylon, unlike PDMS, gains positive charge against copper so that gas breakdown is sustained by secondary electron emission from the copper surface as desired. Details of this test are included in the revised manuscript, including the fabrication of a nylon surface with PDMS foundation and the introduction of a mechanism of friction to enhance charging. From the Paschen curve measured in this test we performed the same step as mentioned in the response to Remark 2 to estimate the secondary-electron-emission coefficient of copper in nitrogen, which is discussed in the response to Remark 4. Meanwhile, we implemented the conventional setup with a high voltage power supply to measure the breakdown voltage of nitrogen both with copper and PDMS-coated aluminum as the cathode, details of which are included in the revised manuscript as well as attached below. These

tests yielded a good match with the Coulomb-force method despite some discrepancies and inconsistencies discussed in the revised manuscript.

Revisions

(Our apologies for the compromise that revisions based on this remark may partly overlap with the previous response to Reviewer 1)

In Results: Nitrogen discharge in copper-nylon electrification

The application of the proposed experimental method on gas breakdown between surfaces with inferior smoothness and compliance faces challenges in the efficiency of charge deposition under contact forces within the safety overload of the load cell for measuring Coulomb forces. This is critical in situations where soft candidates such as PDMS cannot be used as one of the surfaces, e.g., when secondary electron emission from metal surfaces is to be characterized, since most of these surfaces gain positive charge in contact against PDMS. The following demonstrates preliminary strategies on enhancing charge transfer between non-elastomeric surfaces with test results of nitrogen discharge in copper-nylon electrification. Nylon (6-6) is selected as it gains positive charge in contact against copper (the polarity is both inferred by triboelectric series and verified experimentally), so that gas breakdown resulting from copper-nylon electrification is sustained by secondary electron emission from the copper surface. The nylon surface sample (film) is backed with a PDMS foundation (explained in Methods) to improve the effective contact area as well as to avoid plastic deformations during contacts. The contact area remains 45.6 cm² circular and the same pseudo-constant-pressure and -distance test procedures are followed to measure the breakdown voltage of nitrogen. Friction (rubbing) is introduced during the contact cycles (Fig. 5a), where the nylon sample is rotated 57.6° at peak compression force (20 N, 4.4 kPa) and rotated back after the surfaces are fully separated. Results of pseudo-constant-pressure and -distance tests and the collected breakdown voltage measurements are presented in Figs. 5b, 5c and 5d, respectively. It shows that with copper as the negatively charged surface (cathode), gas discharge on the left of the Paschen minimum is comparatively mild, i.e., the decrease of gap voltage at each breakdown event is small. Meanwhile, minor discharge events at low distance-pressure products prior to Paschen predictions are again observed. Energy dependence of the effective secondary-electron-emission coefficient for copper surface in nitrogen breakdown is estimated and shown in Fig. 6 with comparison to that extracted from Paschen curves in literature⁴⁰. Moreover, in copper-nylon contact electrification cycles the van der Waals adhesion is relatively trivial so that pseudo-constant-pressure tests may be initiated when the gap is closed.

(New) Fig. 5 | Nitrogen breakdown in copper-nylon contact electrification. **a** Adapted test apparatus to include surface friction. **b** Pseudo-constant-pressure test results at multiple target gas pressures as labeled with tolerances. **c** Pseudo-constant-distance test results at multiple target gap distances labeled with their nominal values at zero load cell deflection. **d** Collected gap voltage readings at breakdown events detected from test runs in both strategies.

In Results: Supplementary measurements via applied voltage

The presented method aims to directly characterize gas breakdown triggered by triboelectric surface charge. For comparison, nitrogen breakdown is tested in a conventional setup (described in Fig. 7a and Methods) between metal electrodes with an externally applied voltage. Constant-distance tests are performed where breakdown events are recorded at various gas pressures. The breakdown voltage of nitrogen between aluminum electrodes at a gap distance of 1 mm, where the cathode is coated with PDMS, is depicted in Fig. 7c along with comparison to results from Fig. 3c, and that with copper cathode and aluminum anode without coating at a gap distance of 0.8 mm is shown in Fig. 7d along with results from Fig. 5d. Higher

(New) Fig. 7 | Supplementary test with applied voltage. **a** Schematics of test setup. **b** (Left) PDMS(coated cathode)-aluminum(anode) and (right) copper(cathode)-aluminum(anode) setup. **c** Measured breakdown voltage of nitrogen with PDMS-coated cathode. **d** Measured breakdown voltage of nitrogen with copper cathode, with comparison to literature⁴⁰.

breakdown voltages in regions near the Paschen minimum are observed, which is attributed to factors discussed in nitrogen-PDMS-acrylic test results. Deviations may also be due to the current threshold (trip) used for breakdown detection in the high voltage power supply setup so that the Coulomb-force method is in general more sensitive to minor currents, while slight inconsistency in tests with PDMS-coated cathode may be related to charge accumulation on the dielectric surface altering electron states on the (coated) cathode surface and therefore affecting its secondary emission properties.

Remark - 4

4. Alternatively, I suggest that the authors should include at a plot of the secondary emission coefficient for these materials vs. electron energy (either from their own research or values found in the literature) to show that the values they infer with their results are indeed consistent if not similar to those measured independently. Perhaps add a discussion about the energy range in which they match or not. Presently, in this respect they do mention trends, but the plot I suggest would make it more evident for the reader that the secondary emission coefficient is not simply used as a fitting parameter.

Response

We appreciate this suggestion. In the following response we assume that by ‘electron energy’ we are investigating the energy of the incident ions on the negatively charged surface (cathode) during a breakdown event. To avoid potential misunderstanding, we would like to clarify that Fig. 4 in the original manuscript was meant to depict the energy dependence, where we plotted the effective secondary emission coefficient with respect to the reduced electric field, which is theoretically proportional to the energy each cation gains through traveling its mean free path. Unlike secondary emission from incident electron beams, in ion-induced secondary emission the relation between the coefficient and the incident particle energy is determined by both the surface material and the gas constituents. Despite qualitative comparisons with references listed in the response to Remark 2, while we unfortunately do not have available independent information on the secondary electron emission from PDMS under bombardment of nitrogen cations, we were able to draw a comparison between the new copper-nylon test and results from literature¹ on nitrogen breakdown between copper electrodes. This is shown in Fig. 6 in the revised manuscript. The coefficient values show a good match within the energy range covered by the cited work, while our results extended to both lower and higher regions. Nonetheless, the reference values were again extracted from an experimental Paschen curve instead of from independent ion bombardment experiments. (*Casually speaking, same information could be drawn by comparing the Paschen curves directly.*) Yet we hope that this may to some extent serve the purpose of supporting our measurements.

(New) Fig. 6 | Effective secondary-electron-emission coefficient of copper in nitrogen discharge at room temperature 20 °C, estimated from nitrogen breakdown voltage measured between a copper-nylon contact pair, with comparison to reference values extracted from literature⁴⁰.

1. Miller, H. C. Paschen curve in nitrogen. *Journal of Applied Physics* 34, 3418-3418 (1963).

Remark - 5

5) The authors should include a discussion about how the magnitude of the surface charge they infer in the supplemental material compares to that found in the literature.

I recommend the publication of this paper as long as the changes proposed are carried out by the authors.

Response

We appreciate this suggestion and we agree that a comparison of surface charge density measurements can help validate the experimental setup. We assume that our reviewer is referring to results shown in Supplementary (Extended Data) Fig. 3 in the original manuscript. The purpose of this supplementary test conducted in room air is to determine the polarity of surface charge for different combinations of contacting materials, as well as to validate the accuracy of surface charge density measurement using Coulomb force. The actual value of surface charge density shown in this example was both unsaturated and subjected to untraceable breakdown discharge during the transfer of samples in air. Meanwhile, the initial surface charge densities in the gas breakdown tests are also arguably unsaturated, since we made the compromise to use comparatively large samples to guarantee the resolution of force measurements at low breakdown voltages and as a consequence a saturated initial surface charge can easily overload the low-capacity load cell. Therefore, a new separate test was performed to help address this comment, which is described in the revised manuscript. Briefly, we implemented contact electrification between PDMS and a small iron sample (24.3 mm diameter, disc) to approach surface charge saturation under a 20 N peak contact force, which is done in vacuum to avoid gas discharge. An apparently saturated surface charge density of approximately $480 \mu\text{C}/\text{m}^2$ was measured, which is higher than, while comparable in magnitude to, a value of $243 \mu\text{C}/\text{m}^2$ reported in a recent work² for Fe-PDMS contact electrification. (By the way although it might be common knowledge to our experienced readers, we found it intuitively impressive that in vacuum the triboelectric charge on a coin-size surface generated a Coulomb force greater than 5 N)

Again, we sincerely appreciate the reviewer's recommendation and we hope we have addressed the remarks appropriately.

2. Liu, D. et al. Standardized measurement of dielectric materials' intrinsic triboelectric charge density through the suppression of air breakdown. *Nature Communications* 13, 6019 (2022).

Revisions

In *Discussion*: ... A separate test on Fe-PDMS contact electrification with a further reduced effective contact area of 4.62 cm^2 (circular, 24.3 mm diameter) indicates a nearly saturated surface charge density of approximately $480 \mu\text{C}/\text{m}^2$ (Supplementary Fig. 4), which is comparable in magnitude to values reported in literature⁴¹.

(New) Supplementary Fig. 4 | Fe-PDMS contact electrification cycles. **a** Time history of bottom load cell readings, positive being attraction. **b** Time history of top load cell readings, positive being attraction. **c** Surface charge density estimated from top load cell readings. **d** Sample surfaces.

Response to Reviewer #3

General remark

This manuscript should be accepted for publication in Nature Communications. The paper showed direct evidence for Paschen based gas discharge between two polymer plates. The results must be welcomed by the community of triboelectrification.

Response

We appreciate the reviewer's evaluation of the manuscript.

Remark - 1

Just for readability of the document, some minor revision may be suggested:

1. Figure 2 looks a little too busy or noisy. The drawings of electrodes may not necessary.

Response

We appreciate the comment. We agree that Fig. 2 may look crowded with all schematics included, and therefore we made an effort to remove the demonstrative content accordingly while reorganizing the schematics and including them in the Supplementary Information instead. Unfortunately, after the removal of the display items the figure appeared a little too empty instead (attached here). We struggled to choose which version to use but we eventually decided to keep the drawings as we were concerned that general readers who may not be as familiar with the topic as our experienced reviewers might want to use them as reference. Nevertheless, we strived to both simplify and clarify the schematics by removing unnecessary details and using both colors and legends (attached in the following). We truly regret that we were not able to completely fulfill the suggested change and we would appreciate it very much if our reviewer may kindly agree with the current revised form of Fig. 2 and the compromise that we had to take.

Revisions

Please refer to revised Fig. 2 which is also attached in the following.

Remark - 2

2. Presenting some examples of recorded signals of force measurement with gas breakdown events may be helpful to understand the experiments.

Response

We appreciate the suggestion. We agree that the inclusion of the time history of measurements will be helpful to the explanation of the experimental approach, as we realized that both in the presented figures and in the published data we extracted only readings of gas pressure, gap length and force while time stamps were omitted. Hence we included a supplementary figure (attached in the following) where we plotted time traces of the Coulomb force measurement along with those of the respective variables (gas pressure or gap distance) for multiple representative test runs presented in Fig. 3. We hope the raw data may benefit the illustration of the test strategy and more importantly help with the assessment and potential replication and improvement of the present work.

Fig. 2 with drawings removed (backup version)

Revisions

Please refer to new Supplementary Fig. 8 which is also attached in the following.

Remark - 3

3. As discussed in figure 5, there might be a 'jump-off' at the disengaging after Van der Waals adhesion. Please describe the gap distance after the disengaging.

Response

We appreciate the comment, and we are glad that this is brought up. In the separation phase of PDMS-acrylic charging cycles, van der Waals adhesion causes both the elastic deformation of the bulk PDMS material and the deflection of the load cells, which when released result in a non-trivial gap. This may potentially trigger undesired breakdown events in the preparation phase of each test run, as the disengaging process is relatively rapid so that the current data acquisition setup may not have sufficient sampling rate to capture the transient force and gap distance measurements. As described in the original manuscript, in tests starting from the left end of the Paschen curve, this has been avoided by running the charging cycles at sufficiently low gas pressures so that the maximum gap immediately after disengaging remains distant from the breakdown threshold, while in tests starting from the right side of the Paschen curve the compromise is made that the tests are started with a partially dissipated surface charge density. However, if test conditions permit (i.e., in a chamber designed for both vacuum and high pressure), it is also theoretically possible to run the charging cycles at sufficiently *high* gas pressures to avoid such partial discharge.

We appended a supplementary figure (attached in the following) which shows the displacement of the moving surface with respect to the readings of the top load cell (high capacity) during a typical PDMS-acrylic charging phase at low pressure. Immediately after overcoming van der Waals adhesion the displacement reads 1.38 mm (the nominal gap distance), and the true gap distance is slightly smaller when offset of the load cell deflection caused by Coulomb attraction is considered. In fact in the contact cycles the controller is programmed to stop the surface motion at a fixed displacement briefly larger than that at disengaging, in this case 1.5 mm. The presented charging cycles were run at a gas pressure around 6 Pa and, as a conservative reference, the Coulomb-force-displacement relation in the pseudo-constant-pressure test run at around 13 Pa is plotted in the same figure, which shows first discharge events at approximately 2.7 mm and 3.9 mm. Moreover, depending on hysteresis of the surface materials it is also possible to further reduce the gap at disengaging by performing the separation step at a slower rate.

We would also like to mention a new set of tests appended to the revised manuscript, where nitrogen breakdown in copper-nylon electrification is investigated. Compared to tests involving elastomeric surfaces (e.g., PDMS), the quality of contacts is reduced due to lower surface compliance, for which a mechanism of friction is introduced to facilitate surface charge deposition. At the same time, however, the van der Waals adhesion is also reduced and almost trivial in this case, so that the surface may disengage in a smooth manner and the risk of triggering early gas breakdown is trivial. This is shown in the same supplementary figure with the top load cell readings during typical copper-nylon charging cycles.

Lastly, following this discussion we would like to point out observations on the free oscillations of the (bottom) load cell axis which may potentially complicate the test process. Considering its deflection stiffness coupled with inertia of the mounted (bottom) sample, the load cell has a finite settling time and more importantly, a sudden change of load causes it to oscillate at an intrinsic natural frequency. This may be significant in two scenarios: at the moment when van der Waals force is overcome and at the moment of each breakdown event. While the maximum gap distance during such oscillation usually remains safe from

(Revised) Fig. 2 (caption omitted)

(New) Supplementary Fig. 8 | Time histories of Coulomb force measurements in representative test runs from Fig. 3.

triggering gas breakdown in contact cycles, undesired discharge may happen when the load cell axis oscillates due to sudden Coulomb force changes at each breakdown event. Although this does not affect the accuracy of the measurements in general, it may reduce the resolution of the recorded data points for each test run since each breakdown event now dissipates extra surface charge. In other words, it may become a source of error if the amount of surface charge relaxation during each breakdown event is to be quantified using the same test setup. Such vibrations are shown in a new supplementary video using a new Fe-PDMS contact pair, where it also shows that the oscillation during surface disengaging is eliminated when the surface charge density is so high that the Coulomb force itself overloads the load cell readily.

Revisions

Please refer to new Supplementary Movie and new Supplementary Fig. 7 which is also attached in the following.

(New) Supplementary Fig. 7 | Details of charging cycles. **a** Time histories of top load cell readings and top surface displacements in typical PDMS-acrylic charging cycles in 6 Pa nitrogen. **b** Coulomb force measurement in a pseudo-constant-pressure PDMS-acrylic test for nitrogen breakdown at around 13 Pa. **c** Schematics of differences in the magnitude of van der Waals adhesion in the separation phase of PDMS-acrylic and copper-nylon contact cycles. **d** Time history of top load cell readings for typical copper-nylon charging cycles.

Remark - 4

4. Figure 4 could show the recognition between the results of the pseudo-constant-distance tests and the pseudo-constant-pressure tests.

Response

We appreciate the suggestion and have applied colored legends in Fig. 4.

Revisions

Please refer to revised Fig. 4 which is also attached in the following.

Again, we sincerely appreciate the reviewer's recommendation and we hope we have addressed the remarks appropriately in the above.

REVIEWERS' COMMENTS

Reviewer #1 (Remarks to the Author):

The manuscript was revised in accordance with my suggestions. The authors answered the questions given in my last review report. Therefore, I have no more questions and suggestions. I would like to recommend the manuscript be published in its present form.

Reviewer #2 (Remarks to the Author):

The authors have answered satisfactorily all my questions, added relevant references, and made new experiments to validate their results.

I suggest that the manuscript should be accepted for publication in its present form.

Measuring gas discharge in contact electrification

Hongcheng Tao¹, James Gibert¹

¹School of Mechanical Engineering, Purdue University, West Lafayette, IN, USA.

Summary

We (the authors) would like to thank the editor and reviewers for dedicating their valuable time and expertise to assess the manuscript. We appreciate the reviewers' thorough and insightful comments, which significantly contributed to enhancing the quality of the present work. In the following we elucidate our response to each remark, along with the corresponding revisions made in the manuscript, in our pursuit of continued consideration for this submission.

Major changes include:

- A new subsection describing new test results for nitrogen breakdown in copper-nylon electrification
- A new subsection on supplementary test results for gas breakdown using a high voltage power supply
- Supplementary results for charge accumulation in Fe-PDMS electrification
- A supplementary video of load cell deflections and oscillations

The major edits are highlighted in **blue** in a separate copy of the revised manuscript, as well as attached to related reviewer remarks in this response letter for convenience of reference. Additionally, the structure of the main text is briefly adjusted according to formatting instructions of resubmission to *Nature Communications*.

Response to Reviewer #1

General remark

The paper proposes a nondestructive method similar to that reported by other authors in a prior work which uses Coulomb force measurements to monitor surface charge variations and thus quantify the breakdown voltage of the gas between electrified surfaces with respect to its pressure and the gap distance.

The paper really proposes a good method and provides the experimental evidence for a Paschen-type behavior of nitrogen breakdown between a silicone-acrylic pair.

Response

We appreciate the reviewer's evaluation of the present work.

Remark - 1

However, the paper is not recommended for publication in Nature communications for the following two reasons.

First, the similar method had been reported by other authors nearly 30 years ago.

Response

We appreciate the reviewer's evaluation of the experimental approach, and we agree that the method of measuring surface charge density in contact electrification via Coulomb force has been proposed and adopted readily in literature as the physics is clearly fundamental. We would like to clarify that in the original submission we did not intend to claim novelty of the present work on the successful implementation of surface charge density measurement (despite minor contributions in technical details). Instead, we hope we may have the opportunity to explain to our reviewer on the present work's more important contributions both in *applying the Coulomb-force method on the characterization of gas breakdown* and in the *examination/validation of the assumed Paschen curves governing gas breakdown in contact electrification* (discussed in the response to Remark 2). The apparent challenge of achieving these goals lies in acquiring non-trivial measurements of breakdown

voltage near the minimum point on Paschen curves. While there is typically already a significant difference in the magnitudes of the Coulomb force and the contact force required to deposit the surface charge, the Coulomb force corresponding to the lowest breakdown voltages can be even orders of magnitude smaller. For example, in the presented tests of nitrogen breakdown in PDMS-acrylic electrification, the peak contact force during the charging cycles is 20 N, the pre-breakdown Coulomb forces are between 0.1 N to 1N, while the Coulomb forces measured at breakdown events near the Paschen minimum are typically around 0.01N. In particular, we define a major objective of the present work as to show clear evidence that a complete Paschen curve can be obtained with the proposed setup and we demonstrated such ‘completeness’ by showing the connectivity among multiple segments of the Paschen curve recorded in tests under various conditions, as well as by showing a non-trivial residual (survival) surface charge density after the gas pressure is swept over the entire range of interest in pseudo-constant-distance tests. While technically straightforward, we learned that such strategies of reconstructing Paschen curves, as shown in Fig. 2 of the original manuscript, are not currently derivable from literature, and we thus believe that the experimental proof of the engineering feasibility of acquiring complete Paschen curves using the Coulomb-force setup is a novel and fundamental contribution potentially helpful for future related works to refer to.

Remark - 2

Second, such a breakdown obeying Paschen-law is predictable.

Response

We appreciate the comment and we agree that the classical Paschen’s law is generally applicable as a guideline in the modeling and analysis of gas breakdown in contact electrification. It is indeed the fact that Paschen’s law has been assumed so widely in studies concerning gas breakdown triggered by surface charge that motivated the present work to provide an experimental validation for this assumption, for which we learned that insufficient evidence is available from literature so far. As we believe in principle that all theories may benefit from direct experimental examination, we hope we may have the opportunity to explain to our reviewer on the present work’s contributions in and beyond the reconstruction of Paschen curves in contact electrification. As a theory that continues to develop, the formulas of the classical Paschen’s law as well as its various amended versions consist of parameters dependent on properties of both the operating gas and the surface materials. These parameters as well as potential corrections to the conventional theory need to be experimentally determined for each independent scenario, where numerous ongoing efforts are being made in literature to characterize the discharge behavior of different gases between different surfaces (electrodes), despite that in most cases the obedience of classical Paschen’s law is qualitatively predictable. In fact, our test results readily exhibited deviations from predictions of the classical Paschen’s law which is mainly attributed to the energy dependence of the secondary-electron-emission coefficient originally assumed constant in the formula. We were able to extract this information from the measured Paschen curve, as depicted in Fig. 4 of the original manuscript, which matches both in trend and magnitude with reported results from literature on various gases and electrode materials. It is for this same reason that while the rudimentary goal of the present work was to ‘verify Paschen’s law in contact electrification’, we decided to be cautious and precise on the statement and instead described our findings as a Paschen-type behavior (rephrased using ‘Paschen curves’ in the revised manuscript). Based on these results, we attempted to discuss the broader applications of the present work in serving as a preliminary investigation of alternative methods to characterize gas breakdown in general. (*Casually speaking, we would like to explore the possibility of proceeding from ‘measuring gas discharge in contact electrification’ to ‘measuring gas discharge using contact electrification’.*) Despite practical challenges including quality and efficiency of charging between arbitrary surfaces, the presented method may arguably possess potential advantages compared to the conventional setup with a high voltage power supply. For example, in the conventional setup the geometries of the electrodes and the vacuum chamber may require specific designs to avoid undesired discharge paths, e.g., when a gap is formed between two exposed points that is either shorter or longer than the designated one so that it triggers early gas breakdown, while in the Coulomb-force setup with the absence of any circuit or undesired conductive parts the electric field is comparatively localized in the designated clearance gap. Meanwhile, in conventional tests using insulator/dielectric coatings on electrodes, the charge accumulation on the dielectric surfaces resulting from each breakdown event is difficult to either quantify or nullify even with an AC setup, which may affect electron states and thus secondary-emission properties of the surface or potentially break down the dielectric layer itself, while in the Coulomb-force setup since the polarity comes directly from transferred surface charge it is guaranteed that the dielectric surfaces are being neutralized during each breakdown event. We sincerely hope that the above may reassure our reviewer of the necessity and potential applications of the experimental results in the present work.

Remark - 3

I would like to suggest the paper be submitted to Review of Scientific Instruments. Before the submission, it is better to validate the results of the measurement. For example, you may use two plane-parallel metal electrodes with the cathode covered with

the silicone film or the acrylic film. Then, you can measure and compare the breakdown voltages under the same experimental conditions.

Response

We thank the reviewer for the recommended submission to the prestigious journal *Review of Scientific Instruments*. With thorough deliberations, we regret to say we have concluded that the present work may not align with its intended scope optimally. As clarified in our response to Remark 1, we do not claim the current work's contributions on the implementation of an instrument that performs controlled contact electrification and measures surface charge density via Coulomb force. Despite a few novel technical features, we are concerned that excessive emphasis on the design of our experimental apparatus may inevitably overshadow the more significant aspects of the present work. That is, in short, we would like to focus on the proposed methodology of measuring ubiquitous surface-charge-induced gas breakdown, and we believe that the presented experimental evidence of complete Paschen curves reconstructed free of circuits represents a non-trivial addition to the literature. We also believe that interpretations and applications of these results are potentially comprehensive for being useful in various subjects where the characterization and modeling of gas breakdown are concerned, as readily discussed in the manuscript. Therefore, we are honored to take the opportunity to resubmit our revised work to *Nature Communications* in the hope of broadening the exposure of this work to readers from interdisciplinary backgrounds.

We appreciate very much the suggestion of supplementary tests for validation. In the revised manuscript we included results and discussion of breakdown tests on nitrogen with PDMS-coated cathode in a conventional setup utilizing a high voltage power supply. These results yielded a good match with the Coulomb-force approach while sources of deviations and inconsistencies are discussed in the manuscript as well as attached below. Attached below before that is a new set of tests that we appended on nitrogen breakdown in copper-nylon electrification, which we would also like to bring to our reviewer's attention if possible. This is to address the method's applicability on different surface materials so that secondary-electron-emission properties of any material of interest may be investigated by proper choice of the opposite surface and thus the charging polarity. Comparison tests with a power supply are also conducted for this configuration and since copper serves as the cathode we were able to extract reference from literature to provide further validation of our test results.

Revisions

In Results:

Nitrogen discharge in copper-nylon electrification

The application of the proposed experimental method on gas breakdown between surfaces with inferior smoothness and compliance faces challenges in the efficiency of charge deposition under contact forces within the safety overload of the load cell for measuring Coulomb forces. This is critical in situations where soft candidates such as PDMS cannot be used as one of the surfaces, e.g., when secondary electron emission from metal surfaces is to be characterized, since most of these surfaces gain

(New) Fig. 5 | Nitrogen breakdown in copper-nylon contact electrification. **a** Adapted test apparatus to include surface friction. **b** Pseudo-constant-pressure test results at multiple target gas pressures as labeled with tolerances. **c** Pseudo-constant-distance test results at multiple target gap distances labeled with their nominal values at zero load cell deflection. **d** Collected gap voltage readings at breakdown events detected from test runs in both strategies.

positive charge in contact against PDMS. The following demonstrates preliminary strategies on enhancing charge transfer between non-elastomeric surfaces with test results of nitrogen discharge in copper-nylon electrification. Nylon (6-6) is selected as it gains positive charge in contact against copper (the polarity is both inferred by triboelectric series and verified experimentally), so that gas breakdown resulting from copper-nylon electrification is sustained by secondary electron emission from the copper surface. The nylon surface sample (film) is backed with a PDMS foundation (explained in Methods) to improve the effective contact area as well as to avoid plastic deformations during contacts. The contact area remains 45.6 cm² circular and the same pseudo-constant-pressure and -distance test procedures are followed to measure the breakdown voltage of nitrogen. Friction (rubbing) is introduced during the contact cycles (Fig. 5a), where the nylon sample is rotated 57.6° at peak compression force (20 N, 4.4 kPa) and rotated back after the surfaces are fully separated. Results of pseudo-constant-pressure and -distance tests and the collected breakdown voltage measurements are presented in Figs. 5b, 5c and 5d, respectively. It shows that with copper as the negatively charged surface (cathode), gas discharge on the left of the Paschen minimum is comparatively mild, i.e., the decrease of gap voltage at each breakdown event is small. Meanwhile, minor discharge events at low distance-pressure products prior to Paschen predictions are again observed. Energy dependence of the effective secondary-electron-emission coefficient for copper surface in nitrogen breakdown is estimated and shown in Fig. 6 with comparison to that extracted from Paschen curves in literature⁴⁰. Moreover, in copper-nylon contact electrification cycles the van der Waals adhesion is relatively trivial so that pseudo-constant-pressure tests may be initiated when the gap is closed.

(New) Fig. 7 | Supplementary test with applied voltage. **a** Schematics of test setup. **b** (Left) PDMS(coated cathode)-aluminum(anode) and (right) copper(cathode)-aluminum(anode) setup. **c** Measured breakdown voltage of nitrogen with PDMS-coated cathode. **d** Measured breakdown voltage of nitrogen with copper cathode, with comparison to literature⁴⁰.

Supplementary measurements via applied voltage

The presented method aims to directly characterize gas breakdown triggered by triboelectric surface charge. For comparison, nitrogen breakdown is tested in a conventional setup (described in Fig. 7a and Methods) between metal electrodes with an externally applied voltage. Constant-distance tests are performed where breakdown events are recorded at various gas pressures. The breakdown voltage of nitrogen between aluminum electrodes at a gap distance of 1 mm, where the cathode is coated with PDMS, is depicted in Fig. 7c along with comparison to results from Fig. 3c, and that with copper cathode and aluminum anode without coating at a gap distance of 0.8 mm is shown in Fig. 7d along with results from Fig. 5d. Higher breakdown voltages in regions near the Paschen minimum are observed, which is attributed to factors discussed in nitrogen-PDMS-acrylic test results. Deviations may also be due to the current threshold (trip) used for breakdown detection in the high voltage power supply setup so that the Coulomb-force method is in general more sensitive to minor currents, while slight inconsistency in tests with PDMS-coated cathode may be related to charge accumulation on the dielectric surface altering electron states on the (coated) cathode surface and therefore affecting its secondary emission properties.

We hope that in the above we have explained and clarified the methodology, results and contributions of the present work in a better way, and we hope that our reviewer may find our response to the remarks appropriate.

Response to Reviewer #2

General remark

This work presents an investigation on a method to measure gas discharges between insulating bodies charged through contact electrification.

The authors use simultaneous measurements of separation and electrostatic force between the separating bodies at the moment of the discharge to calculate the voltage between them assuming a parallel plate geometry at different gas pressures. The results and analysis are clearly stated, and experiments seem to have been made carefully and methodically.

Response

We appreciate the reviewer's evaluation of the present work.

Remark - 1

1. A very similar methodology was employed in the following work, that I think should be cited:

Collins, Adam L., et al. "Simultaneous measurement of triboelectrification and triboluminescence of crystalline materials." *Review of Scientific Instruments* 89.1 (2018).

Response

We thank the reviewer for the recommended citation. The contributions of the given work and its direct relevance to the present work were recognized upon initial review, but our apologies that its inclusion in the original manuscript was inadvertently omitted. It is cited in the revised manuscript in related statements.

Revisions

In *Introduction*: ... The present work therefore implements an alternative nondestructive approach similar to setups reported in prior works^{7,24} which uses Coulomb force measurements to...

In *Results*: ... Meanwhile, effective charging of harder surfaces using the same test setup may require extra contact force, finer surface topography as well as the introduction of friction²⁴ ...

Remark - 2

2. When compared to both this reference and Horn's 1992 paper (ref. 7), the contribution from the present work should be made absolutely clear:

I strongly recommend this because, at first glance, it seems that the authors are able to reconstruct Paschen's law which would be enough proof of the validity of their results and the novelty of their research. However, as the authors state themselves, they need to assume at least two different values of the secondary electron emission coefficient to reconcile their results with Paschen's law. Furthermore, as they show in Fig. 4, a coefficient for these materials varying by more than four orders of magnitude would be needed. But the authors do not provide independent measurements of this coefficient as a function of electron energy (other than some references). Consequently, it seems that they use the secondary coefficient as a fitting parameter, which would make their results less interesting, specially compared to the works cited above. In this respect, saying that they obtain a "Paschen-type behavior" as they do in the abstract, it really not that surprising.

Response

We appreciate the comment and we agree that contributions of the present work may need further clarification. In particular, we appreciate the reviewer's identification of the current study's novelty in the reconstruction of Paschen curves for contact electrification, and we hope we can have the opportunity to explain our reasoning behind how we presented our work in the original manuscript. We have been aware that a statement that we 'experimentally validated Paschen's law for gas breakdown in contact electrification' would be ideal to conclude the findings, as it was the rudimentary motivation of the present work. However, we took extra caution to be precise about the wording since it has been evident, both in literature and our study, that the classical Paschen's law is typically a general approximation which may require corrections according to various scenarios. A major source of error is indeed the discussed energy-dependent secondary-electron-emission coefficient that is conventionally

assumed constant in Paschen's law. As a consequence, we decided to describe the experimental results as a Paschen-type behavior; which represents a general valley-shaped curve of breakdown voltage vs pressure-gap product. However, we agree with the reviewer that such statements may be vague, and therefore in the revised manuscript we rephrased 'Paschen-type behavior' using 'Paschen curves', which is more recognized while allowing corrections to the classical form. It is for this same reason that we did not attempt to use the entire set of recorded data to fit all parameters in the formula of Paschen's law. Instead, the effective secondary-electron-emission coefficients in Fig. 4 were calculated for each individual data point using this formula assuming constant parameters A and B for the gas. This step has been a common practice in literature for characterizing surface secondary electron emission using Paschen curves examined via various approaches¹⁻³. It is apparent that the energy dependence of the secondary-electron-emission coefficient shown in these works, although measured with different operating gases and cathode materials, is comparable to our results (Fig. 4) in both trend and magnitude. Based on these references we have not been concerned about its value varying by more than four orders of magnitude, as we are confident that such variation does not indicate errors in the tests but is instead a valid estimation of its true energy dependence. Moreover, we believe that this large span of magnitudes of the calculated coefficient may actually reveal the potential advantage of the present approach that it is capable of characterizing secondary electron emission for a comparatively large span of incident particle energy (e.g., compared to individual tests presented in references listed below), i.e., in this case nearly five orders of magnitude as shown on the horizontal axis of Fig. 4. (*Casually speaking, we obtained more than four orders of magnitude of the coefficient partly because we covered nearly five orders of magnitude of incident particle energy.*) We shall further illustrate this point in a more quantitative way in our response to Remarks 3 and 4 below.

References

1. Druyvesteyn, M. J., and Fi M. Penning. "The mechanism of electrical discharges in gases of low pressure." *Reviews of Modern Physics* 12.2 (1940): 87.
2. Auday, G., et al. "Experimental study of the effective secondary emission coefficient for rare gases and copper electrodes." *Journal of applied physics* 83.11 (1998): 5917-5921.
3. Phelps, A. V., and Z. Lj Petrovic. "Cold-cathode discharges and breakdown in argon: surface and gas phase production of secondary electrons." *Plasma Sources Science and Technology* 8.3 (1999): R21.

Related results

Fig. 19, Fig. 20
 Fig. 7, Fig. 8, Fig. 9, Fig. 10
 Fig. 5

Revisions

In Abstract: The present work implements ..., providing experimental evidence of Paschen curves governing nitrogen breakdown in silicone-acrylic and copper-nylon contact electrification.

In Introduction: The present work therefore implements an alternative nondestructive approach similar to setups reported in prior works^{7,24} which ... It is illustrated with the reconstruction of complete Paschen curves for nitrogen breakdown in both silicone-acrylic and copper-nylon contact electrification.

Remark - 3

3. It would be ideal to see in the paper the same experiments between conductors (whose secondary coefficients are better known) to reassure the reader that they experimental setup gives the expected results with conventional materials (like, say, copper). But I understand that in that case an external high-voltage supply would be necessary, along with a fast switch which may not be straightforward to implement.

Response

We appreciate this suggestion and we agree that test results with more sets of materials will definitely improve the completeness of the present work. In the original tests we used the PDMS-acrylic combination mainly to demonstrate the experimental approach's capability to measure gas breakdown between exclusively insulator/dielectric surfaces, in which the mechanical properties of PDMS have greatly eased the charging process. Yet a limitation lies in that PDMS gains negative charge against most materials except few (e.g., PTFE), so that it may not be used when secondary electron emission from the opposite surface (e.g., copper, as suggested) is of interest. Therefore, we conducted a new test for nitrogen breakdown with a copper-nylon contact pair where nylon, unlike PDMS, gains positive charge against copper so that gas breakdown is sustained by secondary electron emission from the copper surface as desired. Details of this test are included in the revised manuscript, including the fabrication of a nylon surface with PDMS foundation and the introduction of a mechanism of friction to enhance charging. From the Paschen curve measured in this test we performed the same step as mentioned in the response to Remark 2 to estimate the secondary-electron-emission coefficient of copper in nitrogen, which is discussed in the response to Remark 4. Meanwhile, we implemented the conventional setup with a high voltage power supply to measure the breakdown voltage of nitrogen both with copper and PDMS-coated aluminum as the cathode, details of which are included in the revised manuscript as well as attached below. These

tests yielded a good match with the Coulomb-force method despite some discrepancies and inconsistencies discussed in the revised manuscript.

Revisions

(Our apologies for the compromise that revisions based on this remark may partly overlap with the previous response to Reviewer 1)

In Results: Nitrogen discharge in copper-nylon electrification

The application of the proposed experimental method on gas breakdown between surfaces with inferior smoothness and compliance faces challenges in the efficiency of charge deposition under contact forces within the safety overload of the load cell for measuring Coulomb forces. This is critical in situations where soft candidates such as PDMS cannot be used as one of the surfaces, e.g., when secondary electron emission from metal surfaces is to be characterized, since most of these surfaces gain positive charge in contact against PDMS. The following demonstrates preliminary strategies on enhancing charge transfer between non-elastomeric surfaces with test results of nitrogen discharge in copper-nylon electrification. Nylon (6-6) is selected as it gains positive charge in contact against copper (the polarity is both inferred by triboelectric series and verified experimentally), so that gas breakdown resulting from copper-nylon electrification is sustained by secondary electron emission from the copper surface. The nylon surface sample (film) is backed with a PDMS foundation (explained in Methods) to improve the effective contact area as well as to avoid plastic deformations during contacts. The contact area remains 45.6 cm² circular and the same pseudo-constant-pressure and -distance test procedures are followed to measure the breakdown voltage of nitrogen. Friction (rubbing) is introduced during the contact cycles (Fig. 5a), where the nylon sample is rotated 57.6° at peak compression force (20 N, 4.4 kPa) and rotated back after the surfaces are fully separated. Results of pseudo-constant-pressure and -distance tests and the collected breakdown voltage measurements are presented in Figs. 5b, 5c and 5d, respectively. It shows that with copper as the negatively charged surface (cathode), gas discharge on the left of the Paschen minimum is comparatively mild, i.e., the decrease of gap voltage at each breakdown event is small. Meanwhile, minor discharge events at low distance-pressure products prior to Paschen predictions are again observed. Energy dependence of the effective secondary-electron-emission coefficient for copper surface in nitrogen breakdown is estimated and shown in Fig. 6 with comparison to that extracted from Paschen curves in literature⁴⁰. Moreover, in copper-nylon contact electrification cycles the van der Waals adhesion is relatively trivial so that pseudo-constant-pressure tests may be initiated when the gap is closed.

(New) Fig. 5 | Nitrogen breakdown in copper-nylon contact electrification. **a** Adapted test apparatus to include surface friction. **b** Pseudo-constant-pressure test results at multiple target gas pressures as labeled with tolerances. **c** Pseudo-constant-distance test results at multiple target gap distances labeled with their nominal values at zero load cell deflection. **d** Collected gap voltage readings at breakdown events detected from test runs in both strategies.

In Results: Supplementary measurements via applied voltage

The presented method aims to directly characterize gas breakdown triggered by triboelectric surface charge. For comparison, nitrogen breakdown is tested in a conventional setup (described in Fig. 7a and Methods) between metal electrodes with an externally applied voltage. Constant-distance tests are performed where breakdown events are recorded at various gas pressures. The breakdown voltage of nitrogen between aluminum electrodes at a gap distance of 1 mm, where the cathode is coated with PDMS, is depicted in Fig. 7c along with comparison to results from Fig. 3c, and that with copper cathode and aluminum anode without coating at a gap distance of 0.8 mm is shown in Fig. 7d along with results from Fig. 5d. Higher

(New) Fig. 7 | Supplementary test with applied voltage. **a** Schematics of test setup. **b** (Left) PDMS(coated cathode)-aluminum(anode) and (right) copper(cathode)-aluminum(anode) setup. **c** Measured breakdown voltage of nitrogen with PDMS-coated cathode. **d** Measured breakdown voltage of nitrogen with copper cathode, with comparison to literature⁴⁰.

breakdown voltages in regions near the Paschen minimum are observed, which is attributed to factors discussed in nitrogen-PDMS-acrylic test results. Deviations may also be due to the current threshold (trip) used for breakdown detection in the high voltage power supply setup so that the Coulomb-force method is in general more sensitive to minor currents, while slight inconsistency in tests with PDMS-coated cathode may be related to charge accumulation on the dielectric surface altering electron states on the (coated) cathode surface and therefore affecting its secondary emission properties.

Remark - 4

4. Alternatively, I suggest that the authors should include at a plot of the secondary emission coefficient for these materials vs. electron energy (either from their own research or values found in the literature) to show that the values they infer with their results are indeed consistent if not similar to those measured independently. Perhaps add a discussion about the energy range in which they match or not. Presently, in this respect they do mention trends, but the plot I suggest would make it more evident for the reader that the secondary emission coefficient is not simply used as a fitting parameter.

Response

We appreciate this suggestion. In the following response we assume that by ‘electron energy’ we are investigating the energy of the incident ions on the negatively charged surface (cathode) during a breakdown event. To avoid potential misunderstanding, we would like to clarify that Fig. 4 in the original manuscript was meant to depict the energy dependence, where we plotted the effective secondary emission coefficient with respect to the reduced electric field, which is theoretically proportional to the energy each cation gains through traveling its mean free path. Unlike secondary emission from incident electron beams, in ion-induced secondary emission the relation between the coefficient and the incident particle energy is determined by both the surface material and the gas constituents. Despite qualitative comparisons with references listed in the response to Remark 2, while we unfortunately do not have available independent information on the secondary electron emission from PDMS under bombardment of nitrogen cations, we were able to draw a comparison between the new copper-nylon test and results from literature¹ on nitrogen breakdown between copper electrodes. This is shown in Fig. 6 in the revised manuscript. The coefficient values show a good match within the energy range covered by the cited work, while our results extended to both lower and higher regions. Nonetheless, the reference values were again extracted from an experimental Paschen curve instead of from independent ion bombardment experiments. (*Casually speaking, same information could be drawn by comparing the Paschen curves directly.*) Yet we hope that this may to some extent serve the purpose of supporting our measurements.

(New) Fig. 6 | Effective secondary-electron-emission coefficient of copper in nitrogen discharge at room temperature 20 °C, estimated from nitrogen breakdown voltage measured between a copper-nylon contact pair, with comparison to reference values extracted from literature⁴⁰.

1. Miller, H. C. Paschen curve in nitrogen. *Journal of Applied Physics* 34, 3418-3418 (1963).

Remark - 5

5) The authors should include a discussion about how the magnitude of the surface charge they infer in the supplemental material compares to that found in the literature.

I recommend the publication of this paper as long as the changes proposed are carried out by the authors.

Response

We appreciate this suggestion and we agree that a comparison of surface charge density measurements can help validate the experimental setup. We assume that our reviewer is referring to results shown in Supplementary (Extended Data) Fig. 3 in the original manuscript. The purpose of this supplementary test conducted in room air is to determine the polarity of surface charge for different combinations of contacting materials, as well as to validate the accuracy of surface charge density measurement using Coulomb force. The actual value of surface charge density shown in this example was both unsaturated and subjected to untraceable breakdown discharge during the transfer of samples in air. Meanwhile, the initial surface charge densities in the gas breakdown tests are also arguably unsaturated, since we made the compromise to use comparatively large samples to guarantee the resolution of force measurements at low breakdown voltages and as a consequence a saturated initial surface charge can easily overload the low-capacity load cell. Therefore, a new separate test was performed to help address this comment, which is described in the revised manuscript. Briefly, we implemented contact electrification between PDMS and a small iron sample (24.3 mm diameter, disc) to approach surface charge saturation under a 20 N peak contact force, which is done in vacuum to avoid gas discharge. An apparently saturated surface charge density of approximately $480 \mu\text{C}/\text{m}^2$ was measured, which is higher than, while comparable in magnitude to, a value of $243 \mu\text{C}/\text{m}^2$ reported in a recent work² for Fe-PDMS contact electrification. (By the way although it might be common knowledge to our experienced readers, we found it intuitively impressive that in vacuum the triboelectric charge on a coin-size surface generated a Coulomb force greater than 5 N)

Again, we sincerely appreciate the reviewer's recommendation and we hope we have addressed the remarks appropriately.

2. Liu, D. et al. Standardized measurement of dielectric materials' intrinsic triboelectric charge density through the suppression of air breakdown. *Nature Communications* 13, 6019 (2022).

Revisions

In *Discussion*: ... A separate test on Fe-PDMS contact electrification with a further reduced effective contact area of 4.62 cm^2 (circular, 24.3 mm diameter) indicates a nearly saturated surface charge density of approximately $480 \mu\text{C}/\text{m}^2$ (Supplementary Fig. 4), which is comparable in magnitude to values reported in literature⁴¹.

(New) Supplementary Fig. 4 | Fe-PDMS contact electrification cycles. **a** Time history of bottom load cell readings, positive being attraction. **b** Time history of top load cell readings, positive being attraction. **c** Surface charge density estimated from top load cell readings. **d** Sample surfaces.

Response to Reviewer #3

General remark

This manuscript should be accepted for publication in Nature Communications. The paper showed direct evidence for Paschen based gas discharge between two polymer plates. The results must be welcomed by the community of triboelectrification.

Response

We appreciate the reviewer's evaluation of the manuscript.

Remark - 1

Just for readability of the document, some minor revision may be suggested:

1. Figure 2 looks a little too busy or noisy. The drawings of electrodes may not necessary.

Response

We appreciate the comment. We agree that Fig. 2 may look crowded with all schematics included, and therefore we made an effort to remove the demonstrative content accordingly while reorganizing the schematics and including them in the Supplementary Information instead. Unfortunately, after the removal of the display items the figure appeared a little too empty instead (attached here). We struggled to choose which version to use but we eventually decided to keep the drawings as we were concerned that general readers who may not be as familiar with the topic as our experienced reviewers might want to use them as reference. Nevertheless, we strived to both simplify and clarify the schematics by removing unnecessary details and using both colors and legends (attached in the following). We truly regret that we were not able to completely fulfill the suggested change and we would appreciate it very much if our reviewer may kindly agree with the current revised form of Fig. 2 and the compromise that we had to take.

Revisions

Please refer to revised Fig. 2 which is also attached in the following.

Remark - 2

2. Presenting some examples of recorded signals of force measurement with gas breakdown events may be helpful to understand the experiments.

Response

We appreciate the suggestion. We agree that the inclusion of the time history of measurements will be helpful to the explanation of the experimental approach, as we realized that both in the presented figures and in the published data we extracted only readings of gas pressure, gap length and force while time stamps were omitted. Hence we included a supplementary figure (attached in the following) where we plotted time traces of the Coulomb force measurement along with those of the respective variables (gas pressure or gap distance) for multiple representative test runs presented in Fig. 3. We hope the raw data may benefit the illustration of the test strategy and more importantly help with the assessment and potential replication and improvement of the present work.

Fig. 2 with drawings removed (backup version)

Revisions

Please refer to new Supplementary Fig. 8 which is also attached in the following.

Remark - 3

3. As discussed in figure 5, there might be a 'jump-off' at the disengaging after Van der Waals adhesion. Please describe the gap distance after the disengaging.

Response

We appreciate the comment, and we are glad that this is brought up. In the separation phase of PDMS-acrylic charging cycles, van der Waals adhesion causes both the elastic deformation of the bulk PDMS material and the deflection of the load cells, which when released result in a non-trivial gap. This may potentially trigger undesired breakdown events in the preparation phase of each test run, as the disengaging process is relatively rapid so that the current data acquisition setup may not have sufficient sampling rate to capture the transient force and gap distance measurements. As described in the original manuscript, in tests starting from the left end of the Paschen curve, this has been avoided by running the charging cycles at sufficiently low gas pressures so that the maximum gap immediately after disengaging remains distant from the breakdown threshold, while in tests starting from the right side of the Paschen curve the compromise is made that the tests are started with a partially dissipated surface charge density. However, if test conditions permit (i.e., in a chamber designed for both vacuum and high pressure), it is also theoretically possible to run the charging cycles at sufficiently *high* gas pressures to avoid such partial discharge.

We appended a supplementary figure (attached in the following) which shows the displacement of the moving surface with respect to the readings of the top load cell (high capacity) during a typical PDMS-acrylic charging phase at low pressure. Immediately after overcoming van der Waals adhesion the displacement reads 1.38 mm (the nominal gap distance), and the true gap distance is slightly smaller when offset of the load cell deflection caused by Coulomb attraction is considered. In fact in the contact cycles the controller is programmed to stop the surface motion at a fixed displacement briefly larger than that at disengaging, in this case 1.5 mm. The presented charging cycles were run at a gas pressure around 6 Pa and, as a conservative reference, the Coulomb-force-displacement relation in the pseudo-constant-pressure test run at around 13 Pa is plotted in the same figure, which shows first discharge events at approximately 2.7 mm and 3.9 mm. Moreover, depending on hysteresis of the surface materials it is also possible to further reduce the gap at disengaging by performing the separation step at a slower rate.

We would also like to mention a new set of tests appended to the revised manuscript, where nitrogen breakdown in copper-nylon electrification is investigated. Compared to tests involving elastomeric surfaces (e.g., PDMS), the quality of contacts is reduced due to lower surface compliance, for which a mechanism of friction is introduced to facilitate surface charge deposition. At the same time, however, the van der Waals adhesion is also reduced and almost trivial in this case, so that the surface may disengage in a smooth manner and the risk of triggering early gas breakdown is trivial. This is shown in the same supplementary figure with the top load cell readings during typical copper-nylon charging cycles.

Lastly, following this discussion we would like to point out observations on the free oscillations of the (bottom) load cell axis which may potentially complicate the test process. Considering its deflection stiffness coupled with inertia of the mounted (bottom) sample, the load cell has a finite settling time and more importantly, a sudden change of load causes it to oscillate at an intrinsic natural frequency. This may be significant in two scenarios: at the moment when van der Waals force is overcome and at the moment of each breakdown event. While the maximum gap distance during such oscillation usually remains safe from

(Revised) Fig. 2 (caption omitted)

(New) Supplementary Fig. 8 | Time histories of Coulomb force measurements in representative test runs from Fig. 3.

triggering gas breakdown in contact cycles, undesired discharge may happen when the load cell axis oscillates due to sudden Coulomb force changes at each breakdown event. Although this does not affect the accuracy of the measurements in general, it may reduce the resolution of the recorded data points for each test run since each breakdown event now dissipates extra surface charge. In other words, it may become a source of error if the amount of surface charge relaxation during each breakdown event is to be quantified using the same test setup. Such vibrations are shown in a new supplementary video using a new Fe-PDMS contact pair, where it also shows that the oscillation during surface disengaging is eliminated when the surface charge density is so high that the Coulomb force itself overloads the load cell readily.

Revisions

Please refer to new Supplementary Movie and new Supplementary Fig. 7 which is also attached in the following.

(New) Supplementary Fig. 7 | Details of charging cycles. **a** Time histories of top load cell readings and top surface displacements in typical PDMS-acrylic charging cycles in 6 Pa nitrogen. **b** Coulomb force measurement in a pseudo-constant-pressure PDMS-acrylic test for nitrogen breakdown at around 13 Pa. **c** Schematics of differences in the magnitude of van der Waals adhesion in the separation phase of PDMS-acrylic and copper-nylon contact cycles. **d** Time history of top load cell readings for typical copper-nylon charging cycles.

Remark - 4

4. Figure 4 could show the recognition between the results of the pseudo-constant-distance tests and the pseudo-constant-pressure tests.

Response

We appreciate the suggestion and have applied colored legends in Fig. 4.

Revisions

Please refer to revised Fig. 4 which is also attached in the following.

Again, we sincerely appreciate the reviewer's recommendation and we hope we have addressed the remarks appropriately in the above.